# Coastal Modelling Environment version 1.0: a framework for integrating landform-specific component models in order to simulate decadal to centennial morphological changes on complex coasts

Andrés Payo[1, 2, 3*], David Favis-Mortlock[1], Mark Dickson[4], Jim W. Hall[1], Martin D. Hurst[3, 5†], Mike J.A. Walkden[6], Ian Townend[7], Matthew C. Ives[1], Robert J. Nicholls[2], Michael A. Ellis[3]

[1]Oxford University Centre for the Environment, South Parks Road, Oxford, OX1 3QY, UK
[2]Faculty of Eng. & the Env. Energy & Climate Change, Southampton Univ., Southampton, SO17 1BJ, UK
[3]British Geological Survey, Keyworth, NG12 5GD, UK (current address) *
[4]School of Env. University of Auckland, 10 Symonds St, Auckland Private Bag 92019, NZ
[5]University of Glasgow, East Quad, Glasgow, G12 8QQ, UK (current address) †
[6]WSP|Parsons Brinckerhoff, Keble House, Southernhay Gardens, Exeter EX1 1NT, UK
[7]National Oceanography Centre, Southampton University, SO14 3ZH UK

*Correspondence to*: Andres Payo (agarcia@bgs.ac.uk)

**Abstract.** The ability to model morphological changes on complex, multi-landform, coasts during decadal to centennial time scales is essential for sustainable coastal management world-wide. One approach involves coupling of landform-specific simulation models (e.g. cliffs, beaches, dunes, estuaries, etc.) that have been independently developed. An alternative, novel, approach explored in this paper is to capture the essential characteristics of the landform-specific models using a common spatial representation within an appropriate software framework. This avoid the problems that result from the model-coupling approach due to between-model differences in the conceptualisations of geometries, volumes and locations of sediment. In the proposed framework, the Coastal Modelling Environment (CoastalME), change in coastal morphology is represented by means of dynamically linked raster and geometrical objects. A grid of raster cells provides the data structure for representing quasi-3D spatial heterogeneity and sediment conservation. Other geometrical objects (lines, areas and volumes) that are consistent with, and derived from, the raster structure represent a library of coastal elements (e.g. shoreline, beach profiles and estuary volumes) as required by different landform-specific models. As a proof-of-concept, we illustrate the capabilities of an initial version of CoastalME by integrating a cliff-beach model and two wave propagation approaches. We verify that CoastalME can reproduce behaviours of the component landform-specific models. Additionally, the integration of these component models within the CoastalME framework reveals behaviours that emerge from the interaction of landforms, which have not previously been captured, such as the influence of the regional bathymetry on the local alongshore sediment transport gradient, the effect on coastal change on an undefended coastal segment and on sediment bypassing of coastal structures.

## 1 Introduction

Coastal managers worldwide must plan for decadal to centennial time horizons [e.g. *Nicholls et al.*, 2012] and may well need to also assess longer-term adaptation measures [*Brown et al.*, 2014; *Hall et al.*, 2012]. However, quantitative prediction of morphological coastal changes at meso-scales (decades to centuries and 10s to 100s of km) is scientifically challenging. Physics-based, reductionist models that represent small-scale processes have proven to be of limited use in this task, both because of the accumulation of small errors over long timescales [*de Vriend et al.*, 1993], because of the omission of processes that govern long-term change [*Murray*, 2007; *Werner*, 2003] and computational limitations [*Daly et al.*, 2015]. Faced with this impasse, coastal geomorphologists have begun to adopt simpler behaviourally based approaches or Large Scale Coastal Behavioural (LSCB) models [*Terwindt and Battjes*, 1990]. LSCB models seek to represent the main physical governing processes at appropriate time and space scales [*Cowell et al.*, 1995; *French et al.*, 2016b; *Murray*, 2013]. Central to these approaches has been selective characterisation of the coastline: thus, we have seen the development of models that simulate the temporal evolution of a range of individual elements of coastal morphology, such as coastal profiles, shorelines or estuary volumes. However, modelling of complex coastlines involving multiple landforms (for example, beaches and tidal inlets) requires consideration of interactions between the component landforms, subject to the principles of mass conservation. This is difficult: modelling these interactions is still not commonplace.

One possible way forward is the development and use of model-to-model interfaces; software wrappers that allow coupling of independently-developed component models [*Moore and Hughes*, 2016; *Sutherland et al.*, 2014]. Significant effort has been oriented in this direction during the last decade, in particular by the Open Modelling Interface (OpenMI) and Community Surface Dynamics Modelling System (CSDMS). OpenMI emerged from the water sector as a way to link existing stand-alone models that were not originally designed to work together [*Gregersen et al.*, 2005], while CSDMS draws on a large pool of well understood open-access models [*Hutton et al.*, 2014]. The promise of OpenMI and CSDMS is to provide a unified system to link various models in order to explore broader system behaviour. However, a range of challenges becomes apparent when linking component models in this way. These include difficulties associated with fully accounting for the cumulative effect of various assumptions made by, and uncertainties in, the constituent models, and non-trivial technical issues concerning variable names and units [*Peckham et al.*, 2013]. Such software-coupling frameworks are themselves agnostic with regard to the spatial structures of component models. This creates a further significant challenge when coupling existing LSCB models due to fundamental between-model differences in the conceptualisations of geometries, volumes and locations of sediment. For example, the Soft Cliff and Platform Evolution [SCAPE, *Walkden and Hall*, 2011] model assumes a beach of finite thickness perched at the top of the bedrock shore profile, while one-line approaches assume infinite beach thickness [*Payo et al.*, 2015]. Similarly, a 2-D estuary model uses the bathymetry to define the form as a continuum, whereas an aggregated model, such as ASMITA [*Stive et al.*, 1997; *Townend et al.*, 2016], uses

only the volume of user defined constituent elements. In this context, coastal modellers need an alternative approach to model integration.

We suggest that integrated modelling must go beyond the software coupling issues that have been the focus of OpenMI and CSDMS. Instead, as argued by *Raper and Livingstone* [1995], integrated modelling should deal more directly with the semantics of the various entities modelled. We propose a way to address this: by means of a modular, object-oriented framework in which these entities are the primary constructs. In other words, the objects that interact within the model framework should correspond to the main real-world constructs considered by coastal scientists and managers. Figure 1 illustrates the modelling approach underpinning the proposed modelling framework; representation of space, and of the changes occurring within its spatial domain, involves both raster (i.e. grid) and vector (i.e. coastline, profile and sediment sharing polygons) representations of spatial objects. This is commonplace in modern GIS packages. What is relatively unusual, however, is that in the proposed framework data is routinely and regularly transformed between these two representations during each time step of a simulation.

In this paper, we provide a detailed description of the proposed Coastal Modelling Environment (CoastalME). We also provide a proof-of-concept illustration of its integrative capacity by unifying independently developed cliff, beach and wave propagation models. Validation of the geomorphological outcomes of model runs against real-world data will be the subject of a future study. This manuscript is organized in six sections. In Section 2, we have outlined the background and rationale for the proposed coastal modelling environment. In Section 3, we explain in detail the proposed framework, including the representation of space and time, inputs and outputs, within time steps main operations, treatment of the domain boundary conditions, implementation and CoastalME modular design. In Section 4, we present some simulation results to illustrate how the different model components integrated in this first composition interact to produce realistic coastal morphological changes and we discuss its advantages and limitations. In Section 5, we summarize the main conclusions. In Section 6, we outline the main websites and weblinks from which the code, the input files used for the test cases and a dedicated wiki-site are available.

## 2 Background and rationale of the proposed coastal modelling environment

### 2.1 Determinants of large scale coastal behaviour

The dynamic behaviour observed in coastal geomorphology is the result of complex feedback relationships linking hydrology, sediment transport and resulting bed evolution, driven by time-variant or stationary boundary conditions and modulated by the underlying geology [*Cowell et al.*, 2003]. While coastal scientists do not have a full understanding of the key processes that control the dynamics of coastal morphology as observed at meso-scales, there are a number of processes that have been consistently identified as important:

i. Gradients in wave-driven alongshore transport, related to coastline shape and wave-incidence, provide the alongshore connectivity between different landform complexes [*Murray et al.*, 2013; *Werner*, 2003].

ii. Sediment sources and sinks in the nearshore system generate an alongshore-propagating curvature and a shoreline change signal. Sources and sinks include human manipulations (localized sources or sinks), river mouths (sources), eroding cliffs (sources), spits that grow into bay or estuary mouths (sinks), and sediment fluxes to or from the continental shelf [*Woodroffe*, 2002].

iii. Wave-shadowing effects from protruding coastline features such as headlands, which tend to create a down-drift zone of diverging alongshore flux and associated shoreline response [*Ells and Murray*, 2012]. These effects give rise to emergent coastline features such as cuspate capes and spits [*Ashton et al.*, 2001; *Ashton and Murray*, 2006], which themselves contribute autogenic wave-shadowing effects and result in shoreline undulations.

iv. Underlying lithology and coastline topography exert significant influence on shoreline change rates, in combination with both alongshore transport gradients and sea-level rise [*Carpenter et al.*, 2015; *Valvo et al.*, 2006; *Walkden and Hall*, 2011].

v. Beach-cliff interactions influence the local cliff and shore platform erosion rate [*Payo et al.*, 2014] and provide a significant sediment source [*Walkden and Dickson*, 2008].

vi. Estuaries and tidal inlets are net sediment importers and/or exporters from/to the open coast [*de Swart and Zimmerman*, 2009] and are controlled by a number of ecological processes [*Friedrichs and Perry*, 2001].

Any model framework which is capable of realistically simulating changes in the morphology of complex coasts during decadal to centennial time periods must (at least) include representations of all the above.

### 2.2 Examples of existing large scale coastal behavioural models: beach, cliff and estuary model

Common amongst most LSCB models is the use of simple geometries to represent complex real-world 3D coastal geomorphology. Profile models, coastline models and volumetric models are the three most-used conceptualisations employed to represent meso-scale coastal morphodynamics [e.g. *de Vriend et al.*, 1993; *Fagherazzi and Overeem*, 2007; *Hanson et al.*, 2003]. These three conceptualisations are, between them, capable of representing a great number of different coastal landforms:

i. Coastal profile models simplify the coastal system to a 2D system (with elevation and cross-shore distance being the two dimensions) that assumes alongshore uniformity [e.g. *Kobayashi*, 2016].

ii. In coastline models, the sand beach morphology is represented by a single contour, and such models are therefore often referred to as “one-line” models [e.g. *Hanson and Kraus*, 2011]

iii. Volumetric models represents the different landforms as sediment sharing entities [e.g. *Stive et al.*, 1997].

The Coastal One-line Vector Evolution (COVE) model is a special case of a ‘one-line’ model designed to handle complex coastline geometries, with high-planform-curvature shorelines [*Hurst et al.*, 2015]. COVE was inspired by the Coastal Evolution Model of *Ashton et al.* [2001] but also includes wave refraction around headlands. In COVE (as in other one-line models), the shoreline is represented by a single line (or contour) that advances or retreats depending on the gradient of alongshore sediment flux. This approach necessarily makes a number of simplifying assumptions to conceptualise the coastline in a way that is consistent with a single-line representation:

i. The cross-shore beach profile is assumed to maintain a constant time-averaged form. This implies that depth contours are shore-parallel, and allows the coast to be represented by a single contour line.

ii. Short-term cross-shore variations due to storms or rip currents are considered temporary perturbations of the long-term trajectory of coastal change (i.e. the shoreface recovers rapidly from storm-driven and tidal-driven cross-shore transport).

iii. Wave action is considered to be the main driver of alongshore sediment transport within the surf zone characterized by the height and angle of incidence of breaking waves. Gradients in alongshore transport therefore dictate whether the shoreline advances or retreats, and whether depositional landforms diffuse, migrate or grow.

A key innovation of COVE is that it uses a local, rather than global, coordinate scheme, enabling coastal cells to take on a variety of polygonal shapes such as triangles and trapezoids [see also *Kaergaard and Fredsoe*, 2013]. The coastline is represented as a series of nodes, each of which is associated with a single polygonal cell; between-cell boundaries are created by the projection of cell boundaries perpendicular to the linking-line between nodes, i.e. approximately normal to the local shoreline orientation. Bulk alongshore sediment flux is driven by the height and incidence angle of breaking waves in each polygon.

The Soft-Cliff and Platform Erosion (SCAPE) model is a time-stepping model of soft-shore recession and morphological change on a profile that is assumed normal to the coastline. It comprises both process descriptions and behaviour-oriented representations. Beach sediment volumes are quantified and conserved, although fine-grained sediments are assumed to be lost from the system (i.e. transported offshore). Sediment is released to the beach through rock erosion and is then moved across- and along-shore. The beach form is assumed to be in a morphological steady state, which is consistent with a one-line model, since its profile is unchanging in time, whilst being translated landward or seaward during the simulation. Alongshore variations in beach volume are captured by the representation of a series of shore-normal profiles. Beach volumes at each shore-normal profile are increased or decreased at each time step by the amount released from the rock to the beach system, and by gradients in alongshore sediment flux, including transport across the littoral boundaries.

Offshore waves in SCAPE and COVE are transformed according to linear wave theory and assuming shore-parallel depth contours with no refraction or loss of energy due to bottom friction. These simplifications are appropriate for gently sloping

bathymetries and low planform-curvature, open coasts but additional modifications are required to account for diffraction and refraction in shadowed regions where these assumptions may not be appropriate. COVE includes simple rules for the diffraction and refraction of waves when the coast is shadowed from incoming waves. An alternative model that includes energy dissipation due to wave breaking and bottom friction (while assuming alongshore-uniformity) is the Cross Shore Model [CSHORE, *Kobayashi*, 2016]. CSHORE solves a combined wave and current model based on time-averaged continuity, cross-shore and longshore momentum, wave energy or action, and roller energy equation to estimate wave induced hydrodynamic.

The Aggregated Scale Morphological Interaction between Inlets and Adjacent Coast (ASMITA) model is a behaviour-oriented model that describes the evolution of a tidal inlet towards an equilibrium which is forced by external conditions and geometrically constrained by human interventions [*Stive et al.*, 1997; *Townend et al.*, 2016]. The ASMITA concept has been applied to simulate the effects of the closure of tidal basins, dredging and dumping of sediment, and sea-level rise, on both hypothetical and real tidal basins [*Rossington et al.*, 2011; *Van Goor et al.*, 2003]. ASMITA conceptualizes the estuary as a highly schematised representation of geomorphic elements, for example the ebb-tidal delta, sub-tidal channel, and intertidal flats (as found on barrier island coasts such as the Wadden Sea). The most important assumption underpinning the ASMITA conceptualisation is that a morphological equilibrium for each estuary element is a function of the controlling hydrodynamic (e.g. tidal prism, tidal range) and morphometric (e.g. basin area) conditions. The tidal system can thus be schematized as one, two or three sediment-sharing elements involving the ebb-tidal delta, channel and tidal flat. Ebb-tidal deltas are important sediment reservoirs that may supply the tidal system with sediment, unless the delta is sediment-starved, in which case the system may demand sediments from the adjacent coast. The volume of an ebb-tidal delta can be defined assuming the coast is undisturbed by a coastal inlet, and therefore its bathymetry is assumed equal to that of the adjacent barrier coast. Thus in ASMITA, the volume of the ebb-tidal delta is equal to the volume above this virtual no-inlet coast.

### 2.3 Rationale for a new approach to model large scale coastal behaviour

The three LSCB models outlined above each have different sediment conservation and morphological updating principles, each operates on a different abstraction of coastal geometry, and each uses different sediment accounting structures. However, they also possess some salient attributes, which potentially provide a basis for a shared, generic, geometric and sediment budget-modelling framework. Considering these three models, and LSCB models in general, we observe that:

- All meso-scale models conserve sediment volume/mass.
- Sediment is stored as deposited material (gravel, sand, fine) or held in suspension.
- LSCB models typically employ some characterisation of hydrodynamic forcing (e.g breaking wave height and direction, 1D-estuary water levels and tidal flows, fetch-limited estuary wave’s heights).

- Sediment accounting is on a two-dimensional horizontal grid (2DH) (e.g. TIN, regular, curvilinear, quad-tree, raster); 1D geometries (e.g. shore profiles or a one-line model) may be represented with a 2DH;
- Behavioural models operate on some abstraction of a full 3D topography/bathymetry (e.g. shorelines, shore profiles, sandbank/delta volumes, estuary volumes/cross sections, estuary channel networks, mudflat areas), and appropriately make some classification of the modelled landforms (e.g. one-line models apply to curving sandy rich coastlines, SCAPE models apply to shore profiles);

Thus, we suggest that these three LSCB models can, in common with other LSCB models, be integrated within a modelling framework that respects and emphasizes the above-listed attributes. In the next section, we describe the initial implementation of such a framework.

## 3 Description of the proposed framework

We next present a detailed description of the CoastalME framework objects and methods and how they operate together to capture a generic morphodynamic feedback loop. To demonstrate the integration capacity of the proposed framework we also describe how two landform specific models (COVE for sediment rich open beaches and SCAPE for soft cliffed open coast) can be integrated, combining the simple diffraction and refraction rules used in COVE's wave propagation module and the less restrictive CSHORE wave propagation module. In choosing component models, our aim was to demonstrate how the CoastalME framework permits distinct but morphologically linked processes to be represented in a consistent manner.

CoastalME is not a simulation model but a framework to integrate different model components and therefore, most of the classes and methods that are likely to be modified by a coastal modeller using CoastalME can be simply replaced by an overloading method or class. Some concepts of the model components described above are hard-coded into CoastalME (i.e. are unlikely to be changed by the modeller). We provide more details about the modularity of CoastalME at the end of this section but, the most salient example of a hard-coded concept is the use of simple polygons, like in COVE, to calculate the alongshore sediment transport. Gradients in wave-driven alongshore transport, related to coastline shape, provide the alongshore connectivity between different landform complexes. The use of simple triangular and trapezoidal shapes can accommodate very straight coast as well as highly irregular coastlines ensuring that alongshore connectivity is well captured. Figure 2 illustrates how these polygons looks like on real coastal stretch (Benacre Ness on the East coast of the UK). The length of the coastline-normal used to define the polygon boundaries for this example are 4km perpendicular from the local coastline orientation. Polygons are trapezoidal in shape where the coastline-normals do not intersect and triangular where they do intersect. Sediment is bypassed between polygons with shared boundaries. How the sediment is being eroded and deposited within polygons and shared between polygons is explained below. Figure 2 also illustrate how the bottom

topography might be composed of consolidated rock (Figure 2, profile near Ness) or made off a small layer of un-consolidated material on top of a consolidated profile (Figure 2, profile over Ness). We propose a simple block-data structure to capture both the topography and the stratigraphy.

### 3.1 Inputs, outputs, sediment blocks and conventions

Input parameters for CoastalME are supplied via a set of raster files, and a text-format configuration file. CoastalME's output consists of GIS layer snapshots, a text file, and a number of time-series files. The GIS files include both raster layers such as Digital Elevation Models (DEMs) and sediment thickness, and vector layers such as the coastline. Optionally, there is also the ability to output snapshots of individual geometrical objects such as the shore profile.

Figure 3 illustrates the model's block data structure used to represent the topography and stratigraphy. The smallest spatial scale within CoastalME is a block; blocks are square in plan-view and of variable thickness. A coastal stretch is characterized by a minimum of two raster input files. These are (i) a basement file giving the elevation of non-erodible rock that underlies (ii) a single sediment layer giving the thickness of a single sediment size fraction, either consolidated or unconsolidated. More sediment layers, representing other sediment size fractions (both consolidated and unconsolidated) may be specified if desired. Whilst the basement is a non-erodible layer, consolidated and unconsolidated sediment layers may increase or decrease their thickness during a simulation. Each sediment layer potentially comprises three size fractions, fines (mud/silt), sand (0.063mm < $D_{50}$ < 2.0 mm) and coarse (2.0mm < $D_{50}$ < 63.0mm) sediment; however, any of these size fractions may be omitted for some or all raster cells, in which case the model will assume zero thickness of this size fraction for that raster cell. Non-consolidated layers are assumed to lie above non-consolidated sediment layers, and the size fractions within a layer are assumed to be well mixed within the layer. Consolidated sediments are essentially erodible solid rock while unconsolidated sediments are loose materials, ranging from clay to sand to gravel. Sediment grain sizes for both consolidated and unconsolidated layers are specified in the configuration file (the default assumed values of different fractions sediment size is 0.065 mm, 0.42 mm and 19.0mm for fine, sand and coarse respectively). The sediment mass transferability between these six different types of sediment (three sediment size fractions and two consolidation states) is hard-coded within the CoastalME framework. Consolidated coarse and sand sediment fractions, when eroded, are assumed to become part of the unconsolidated coarse and sand material. In the present version of the model, eroded fine material is simply assumed (as in SCAPE) to become part of a global suspended sediment fraction (i.e. not lost but stored as suspended sediment). The elevation of the sediment top elevation or DEM is obtained by adding together the thickness of all the different layers to the elevation of the basement.

Two files are required to represent structural human interventions such as groins and breakwaters. One raster file represents the thickness of the intervention above the ground while a second raster file represents the class of human intervention (null if no intervention is present and 1 if a structural intervention is present on a given cell of the raster grid).

In CoastalME, we use the International System of Units and the convention for wave direction is the 'true north-based azimuthal system'. This is the oceanographic convention in which zero indicates that the waves are propagating towards the north, and 90° indicates that waves are propagating towards the east. Wave forcing is described by the deep-water properties of incoming waves (significant wave height, peak period and direction) i.e. as unaffected by shallow-water refraction, shoaling and shadowing. Water depth is measured relative to the Still Water Level (SWL); defined as the elevation of the sea surface in the absence of wind, waves and tides. Shoreline orientation is also measured clockwise relative to the azimuth following the convention shown in Figure 4, with shoreline orientation being 0° when oriented S-N and 90° when oriented W-E.

## 3.2 Within time step data flow and operations

CoastalME uses an implicit method (i.e. find a solution by solving an equation involving both the current state of the system and the later one) to iteratively erode and deposit the different sediment fractions over the entire model grid. There is a fixed time step that could be, in principle, of any duration (i.e. hours, days, months, years). In practice, however, there are constraints on time step duration due to the amount of sediment that can be eroded or deposited on a single time step without unrealistically de-coupling the morphology change and assumed hydrodynamic forcing during a given time step [*Ranasinghe et al.*, 2011]. Table 1 summarises the raster-to-vector transformations that take place during a single CoastalME time step: we describe each one in detail below.

### 3.2.1 Non-landform-specific operations

At the beginning of each time step a set of non-landform-specific operations are executed to update the external forcing, trace the coastline and map all landforms types.

External forcing values modify the SWL and the deep-water properties of incoming waves (significant wave height, peak period and direction). In the present version of the framework, SWL may be fixed or it can change linearly every time step, so at the end of the simulated period the user-defined sea level change is achieved (i.e. SWL curve is defined by the initial SWL, duration of the simulation and the SWL at the end of the simulation).

Next, the CoastalME framework traces the coastline on the raster grid by finding the intersection of the ground elevation and the current SWL. For this, we use the well-known wall-follower algorithm [*Sedgewick*, 2002]. Unlike other LSCB models, such as one-contour models, CoastalME does not require the user to define the shoreline location at the initial time step: the framework determines this. Raster cells 'on' the shoreline are marked; the coastline is also stored as a vector object made up of a set of consecutive points, where each coastline point has an associated location (x, y) and elevation (z=SWL+ tide) in the global geographic reference system. Each shoreline point also holds local attributes such as curvature and orientation relative to the azimuth. The inevitable angularity of raster-traced 'lines' means that smoothing of the raster-traced coastline is necessary to remove unrealistic jaggedness. For this, the user may choose to use either a simple moving average or Savitzky-Golay smoothing (fitting successive sub-sets of adjacent data points with a low-degree polynomial by the method of linear least squares: *Savitzky and Golay* [1964]).

The framework classifies each grid cell as a member of a given coastal landform. Of course, such an approach requires a consistent ontology. Here, we have adopted the ontology suggested by [*French et al.*, 2016a], which includes both human interventions (structural and non-structural) and natural landform components (Table 2). For this first version of CoastalME, we have selected a subset of these landforms to illustrate how they can be incorporated into the framework. First all grid cells associated with a human intervention (user-defined) are marked and stored as an intervention vector object. Structural interventions are assumed non-erodible. At the first time step, the framework traverse the coastline cells and marks them as cliff or drift type if the sediment-top-elevation material is consolidated or unconsolidated respectively. If a cliff cell is identified along the coastline, it creates a cliff object with the default cliff properties (i.e. notch overhang, notch base level, accumulated wave energy, remaining cliff to be eroded). On successive time steps these cliff properties will be modified accordingly to the user specified cliff erosion rules. A cliff cell that is transformed into a drift cell at the next time step is classified as an eroding coastal cliff. Other cells are marked as hinterland, non-coastal-cliff, or sea cells, but no landform object has been associated to them yet. Figure 5a illustrate the concepts of raster and vector coastlines as well as landform type classification. In COVE, the coastline is made of a relatively small number of discrete nodes while in CoastalME the coastline is made of a considerably greater number of coastline cells. Therefore, in CoastalME there are many more coastal points between two polygon boundaries than in COVE. CoastalME uses a smoothed vector coastline to trace the coastline-normal. This smoothed coastline is conceptually equivalent to the use of adjacent nodes in COVE.

### 3.2.1 Construction of coastline normals and sediment sharing polygons

Once a coastal stretch has been classified as open coast, a set of sediment sharing polygons are traced using a combination of raster and vector geometries. First coastline-normal profile objects are created (Fig. 5a). Each coastline-normal profile object is equivalent to a shore-normal elevation profile, being a vector line made up of a set of consecutive points where each point has an associated location in the global geographic reference system. The elevation of each point of the coastline normal is

then determined using the elevation of the centroid of the closest raster cell. Figure 5a shows the relationship between the coastline normal in vector and raster format. Elevation values derived directly from rasters can be unrealistically jagged, so (as with the grid-traced coastline) some smoothing of the raster-derived elevation profiles is necessary to give realistic point-to-point gradients along the profile. Each profile point also holds other local attributes such as landward-marching gradient (i.e. the slope of the profile as we move from the seaward limit towards the landward limit). The landward limit of each coastline normal is the centroid location of the cell that has been marked as a coastline cell (i.e. where the profile elevation intersects the SWL). The seaward limit of the profile is, in the present version of the CoastalME framework, a user-defined elevation value: it must, however, be deeper than the depth of closure (i.e. the sea depth beyond which no significant erosional change to unconsolidated sediments is expected). The depth of closure ($d_L$ [L]) is calculated using the empirical expression (1) proposed by [*Hallermeier*, 1978] where $H_{sx}$ [L] is the nearshore storm wave height that is exceeded only 12 hours each year and $T_{sx}$ [T] is the associated wave period and $g$ [LT$^{-2}$] is the acceleration due to gravity. For each time step, changes in the unconsolidated profile are assumed to occur between the landward and seaward profile limits.

$$d_L = 2.28H_{sx} - 68.5\frac{H_{sx}^2}{T_{sx}^2} \quad (1)$$

The along-coast, planform, spacing of these coastline-normal profiles is (in the current version of the framework) approximately specified by the user. However, this spacing is modulated both by coastline curvature and (optionally) by a random factor. This random component of profile spacing means that profiles shift somewhat from time step to time step; this aims to allow the user to explore the impact of any artefacts resulting from profile location. The framework also preferentially locates coastline-normal profiles on 'capes' (portions of the coastline with maximal convex curvature), and preferentially does not locate profiles in small and tight bays (portions of the coastline with maximally concave curvature). Additionally, there is a constraint on the user-defined profile spacing: if this is too small relative to the raster cell size, profiles will very frequently intersect. To avoid this issue, we require the user-defined profile spacing to be more than ten times the cell size. For a typical raster cell size of 5 m, the minimum distance allotted between profiles will be 50 m (which is like the smallest distance recommended by *Hurst et al.* [2015] for COVE). Where coastlines meet the edges of the raster grid, extra profiles are added which follow the grid edge, and are not usually (unless the coastline intersects the grid edge at a right angle) normal to the coastline.

All pairs of coastline-normal profiles are then checked for intersection. If any two profiles intersect then they are merged seaward of the point of intersection, with a planform orientation which is the mean of the two profile orientations. This is necessary because the coastline-normal profiles also serve as boundaries between coastal polygons (see below). The process is repeated until no two coastline-normal profiles cross, prioritising merging of interceptions nearest to the coastline. Figure 5b shows the coastline normals traced along a coastline stretch with a vertical groin interrupting the alongshore sediment transport. Coastline normal far from the intervention do not intersect, while those close to the intervention intersect and are merged as described above. Coastal polygons are thus created using the coastline normals as inter-polygon boundaries.

Coastal polygons in CoastalME are broadly similar to their equivalents in COVE. However, in COVE the inter-polygon boundaries are determined using a linking line which connects up-coast and down-coast nodes, whereas in CoastalME the coastal normals (and hence the inter-polygon boundaries) are constructed as from a larger number of coastal grid cells.

Polygons at the boundaries of the DEM are constructed differently as described above. Boundary conditions are invariably a problem for simulation models [*Favis-Mortlock*, 2013a] and CoastalME is no exception. Profiles at the start and end of the coastline vector are assumed to project along the main intersecting global axis rather than being normal to the coastline location. This is needed to avoid profiles at the edges moving out of the raster grid domain when projected seaward. To specify the sediment fluxes coming in and out of the polygons with boundaries intersecting or at the edge of the raster grid domain the user can select among three types of boundary conditions: (1) an open boundary condition, which permits export of sediment at all grid edges, (2) a closed boundary condition, which assumes that no sediment enters or leaves the raster grid, and (3) a periodic boundary condition for which sediment exported from one end of a coastline is re-imported at the other end of the coastline. The first option permits net loss of sediment from the grid, while the other two options do not. For simulations where wave direction produces a net up-drift or down-drift alongshore movement of unconsolidated sediment, the open boundary option gradually leads to impoverishment of, even total removal of, unconsolidated sediment at the up-drift end of the coast, whereas the closed boundary option eventually leads to an accumulation of sediment at the down-drift end of the coast.

### 3.2.2 Wave transformation

The next step is to propagate the user input wave conditions from deep water to breaking for each raster grid cell and to store a representative set of wave properties both at every point along the coastline, and for every sea cell.

Wave energy flux – the main driver of cliff and shore platform erosion and alongshore sediment transport – can be characterized by the wave height, period and angle at breaking. The CoastalME framework permits wave propagation to be calculated either using the current DEM (i.e. as in many coastal area models), or by assuming a simplified bathymetry (e.g. bottom contours parallel to the shoreline). The current version of the CoastalME framework assumes alongshore uniformity to calculate wave refraction (i.e. application of Snell's Law), simple rules to estimate diffraction as described by *Hurst et al.* [2015] and two different approaches to calculate wave transformation due to shoaling and energy dissipation. The first approach is based on linear wave theory and assumes no energy losses due to wave breaking or bottom friction, in a manner equivalent to the method used by *Hurst et al.* [2015] in COVE. The second approach uses the CSHORE wave propagation module which include energy dissipation due to wave breaking and bottom friction [*Kobayashi*, 2016]. To illustrate the modularity of CoastalME, these two approaches have been integrated into CoastalME as separate methods that can be

selected by the user in the input configuration file. The integration approach is also different; the simpler COVE approach has been fully coded as a new method while the more complex CSHORE approach is called as an external library.

Both wave transformation approaches involve the calculation of wave attributes along each coastline normal: this enables us to identify the depth of breaking and the extent of the surf zone (i.e. the area where waves are breaking) for each coastline-normal profile. Both approaches uses a constant wave height to water depth at breaking ratio (0.78) to assess if waves are breaking. Because CSHORE assumes irregular waves (i.e. instead of COVE's monochromatic waves), the breaking depth is further defined as the depth at which 98% of the waves are breaking. Values for wave attributes (wave height and wave direction) for cells between normals are then interpolated (using GDAL's 'linear' Delaunay triangulation-based method: GDAL, 2017) to all other near-coast cells of the raster grid. Incoming waves are decomposed into their global *x* and *y* components, each component is interpolated and the resulting interpolated wave height and direction is obtained from the interpolated components. A nearest neighbourhood interpolation is used to identify the cells that are in surf zone.

Figure 6 shows the wave height distribution for different incoming wave direction (315°, 270° and 225°) over a gently sloping bay shaped bathymetry. The initial DEM is made of 1m regular square cells that extent an area of 1000m x 500m and has an average slope seaward of 10.25° that varies (±0.25°) alongshore to represent a cuspate coastline. Coastline normals of 280 m length do not intersect for the minimum spacing allotted (50 m for cell size of 1 m). The width of the surf zone is of the order of 15m and shorter than the profile length. Due to refraction, the wave height at the capes is larger relative to the wave height in the bays. A comparison of the across-shore wave height distribution calculated using COVE's and CSHORE's wave propagation approaches is shown in Figure 6e. For this gently sloping bathymetry the across-shore variation of wave height is similar for both approaches; a slight decrease in wave height (deep water wave height is 2 m) as the waves propagates towards the coast followed by a rapid increase of wave height before breaking. For the example shown in Figure 6e, the wave height at the seaward end of the transect for the COVE approach is 1% smaller than the wave height estimated by CSHORE. This difference is due to the way that COVE and CSHORE solves the dispersion relation for linear waves; COVE uses *Fenton and McKee* [1990] approximation while CSHORE solves the dispersion relation iteratively to obtain the wave length and wave frequency at a given depth.

We use the simple rules approach described by *Hurst et al.* [2015] to modify the wave height and direction due to diffraction of waves in the shadow zone. CoastalME traverses all coastline cells searching for shadow zones. Starting with any capes, shadow zones are traced by projecting a straight line parallel to the deep-water wave direction until either another coastal cell or grid edge is hit. Figure 7a shows an example of a projected straight line that crosses another coastal cell created by a groin for a 225° incoming deep water wave-direction defining the shadow zone sea boundary. If the length of the shadow zone boundary is too short the effect on wave transformation is considered negligible. If the length of the shadow zone is long enough (less than 10 m), all the coastline cells under this shadow zone boundary are marked as cells within a shadow zone.

CoastalME also find out if a shadow zone is nested within a larger shadow zone and keeps the largest. Sea cells under the shadow zone and on the area affected by the shadow zone are marked (Figure 7b) and wave attributes modified accordingly. For each cell within the shadow zone, the non-diffracted wave angle, $\theta$, is adjusted as:

$$\theta_s = \theta + 1.5\omega \quad (2)$$

where ω [°] is the angle between the shadow zone boundary line and the line that connect the cape coastline cell and this cell. Diffraction rotates the incoming waves toward the coast, increasing the wave angle if the shadow zone is on the left of the incoming waves and a decreasing it if the shadow zone is on the right. For ω equal or larger than 90°, breaking waves heights are assumed to be zero. The approaching wave height within the shadow zone is calculated by modifying the refracted and shoaled wave height using a diffraction coefficient $K_D$ [dimensionless] which is a function of $\omega$:

$$K_D = 0.5cos\omega \quad (3)$$

In order that wave energy is conserved, the length of the coast in the shadow zone $l_s$ [L] is determined and wave energy is also reduced downdrift of the shadow zone over the same distance. While the alongshore distance back to the shadow zone $x_s < l_s$, where $x_s = 0$ at the tip of the shadow zone (i.e., where $\omega = 0°$; see Figure 7a), wave energy outside the shadowed zone is reduced following:

$$K_D = \frac{1}{2}\left(1 - sin\left[90\frac{x_s}{l_s}\right]\right) \quad (4)$$

Using the modified diffracted wave height, all coastline normal within the shadow zone are traversed calculating the new breaking wave height and breaking water depth using the wave height to water depth ratio as before. Once the wave properties are estimated for all raster grid cells and coastline profiles, including the modification due to wave shadowing, CoastalME then calculates the wave height and angle at breaking for each point along the coastline (Figure 7c). The wave properties at breaking are stored for each coastline cell for later use by the shore platform and cliff erosion and alongshore sediment transport modules. Consolidated cliff and shore platform morphology changes slower than unconsolidated drift material so it is calculated next.

### 3.2.3 Down-wearing erosion of the consolidated shore platform

The current version of CoastalME integrates a slightly modified implementation of SCAPE for erosion of the submerged consolidated profile. First, potential down-wearing erosion (unconstrained by sediment availability) is calculated at every coastline-normal profile (Figure 8a). Potential down-wearing erosion is defined as the maximum erosion estimated to occur during the time step for a given breaking wave height and angle. The horizontal recession at a given shore platform elevation ($z_s$) is calculated in SCAPE by:

$$\frac{dy_s(z_s)}{dt} = \frac{F}{R} f_1 f_2 \left(\frac{dz_s}{dy_s}\right) \quad (5)$$

where shore platform horizontal and vertical dimensions are $y_s$ [L] and $z_s$ [L], respectively; $t$[T] is time and $F = H_b^{13/4} T_p^{3/2}$ [$L^{13/4}T^{3/2}$] is the erosive forces under random waves [*Kamphuis*, 1987] of a given wave height at breaking $H_b$ [L] in meters and peak period $T_p$ [T] in seconds; $R$ [$M^{9/4}T^{2/3}$] is a calibration parameter that varies with the material strength and some hydrodynamic constant in $m^{(9/4)}s^{(2/3)}$; $f_1$[dimensionless] is a shape function that describes how the erosive forces $F$ varies with water depth $(h(t) - z)$ (Fig. 8b), with $h(t)$ [L] equal to the changes in the SWL at a given time $t$; $\frac{dz}{dy_s}$ [dimensionless] is the local slope at the platform elevation z; $f_2$ [dimensionless] is a discontinuous function that is equal to 0 if the beach thickness (the difference between the elevation of the beach and the elevation of the consolidated platform $z_{beach} - z_s$) is larger than $0.23H_b$ and increases linearly up to 1 if there is no beach on top of the shore platform (Fig. 8c). Thus, as the beach becomes thicker, erosion of the shore platform is reduced. For submerged blocks (i.e. blocks for which the top elevation is below SWL), the original horizontal SCAPE erosion (Eq. 5) is converted into its vertical component, $\frac{dz}{dt}$, by applying the following simple trigonometrical conversion;

$$\frac{dz_s}{dt} = \frac{dy_s(z_s)}{dt}\left(\frac{dz_s}{dy_s}\right) \tag{6}$$

All parameters in Eq. (6) are either readily available or can be derived from existing CoastalME parameters. The profile local slope, $\frac{dz_s}{dy_s}$, can be derived from the coastline profiles. For each coastline-normal profile, the erosive force is calculated as a function of the wave height at breaking (stored in the coastline object) and the wave peak period. The same shape erosion function as used by SCAPE is used to estimate the shore platform erosion as a function of the ratio of water depth to wave height at breaking. The beach thickness is calculated for each cell along the profile as the elevation difference between the beach surface elevation and the consolidated bedrock surface elevation at each cell. When a value for potential platform erosion (i.e. not considering the availability of sediment) has been calculated for all points on every coastline-normal profile, we then interpolate these values to the near-coast raster cells by creating a series of temporary elevation profiles, each of which is planform parallel to the up-coast or the down-coast coastline-normal profile (Fig. 8e).

Actual (i.e. supply-limited) values for platform erosion at each raster cell is constrained in CoastalME by the amount of sediment and its availability -how much can be mobilized on each time step or active layer-. SCAPE assumes two sediment fractions with two different behaviours; fine sediment that when eroded is lost as suspended sediment and coarse sediment that becomes part of the drift material (i.e. increase beach volume). CoastalME include three sediment fractions (fine, sand and coarse) and the percentage of each fraction is determined by the thickness of each fraction each raster cell -equivalent as in SCAPE-. The concept of availability factor is included in CoastalME because it is required as soon as erosion rates are managed separately for the different sediment fractions –to include the interaction between the different sediment fractions- [*Le Hir et al.*, 2011]. CoastalME uses an availability factor $\alpha_i$ [dimensionless] (0 for none to 1 for all sediment available) for each one of the three sediment fractions (coarse, sand and fine, denoted by subscript $i$). The actual total erosion for each time step is calculated by:

$$\frac{dz}{dt} = \min(\sum_i \epsilon_i \alpha_i, \sum_i z_i) \quad (7)$$

where $\epsilon_i$ [L] is the potential erosion for sediment fraction *i* and z is the thickness of each sediment fraction. This restriction is applied, to acknowledge that fine sediments are more likely to be eroded in larger amounts than coarser sediment. Eroded consolidated coarse and sand fractions are transferred to the unconsolidated coarse and sand fractions (i.e. as a local source of sediment), adding to the total beach unconsolidated material. Eroded consolidated fine sediment is assumed in the current version of CoastalME (as in SCAPE) to become part of the suspended sediment mass fraction (i.e. do not add volume to the drift material on the beach).

### 3.2.4 Cliff erosion

In SCAPE, an initially uniform slope under the attack of breaking waves starts developing a cliff notch somewhere in between the high and low tidal levels. After a user-defined number of erosive events SCAPE assumes that any overhanging material is removed (i.e. the cliff collapses) which produces a vertical cliff starting from the most landward location of the notch: in other words, the profile shifts in a shoreward direction. In CoastalME a cliff is represented by cells that cannot shift: they can only change elevation. This necessitates a modification of the SCAPE approach, as described below. As in SCAPE, we assume that the cliff and shore platform can only be eroded (i.e. no creation of new consolidated platform is allowed).

In the CoastalME framework, wave energy is accumulated at every point on the coastline: for cliff objects, this results in the development of a cliff notch which is also represented in the raster cell which is associated with this coastline point. The base of the cliff notch is at a user-specified depth $d_1$ below the current water level (i.e. SWL + tide), and the notch is considered to be eroded a length $L_1$ inland (Figure 9a). Eroded coarse/sand-sized sediment fraction is deposited as unconsolidated talus and beach material, and any eroded fine-sized sediment is just added to the accumulated total suspended sediment fraction. The elevation of the talus at its coastward end is set as a user-specified fraction of the cliff height, measured from the notch base. The talus width, in plan view, is also a user-specified value. The beach profile is, in the present version of the framework, assumed to be a Dean profile [*Dean*, 1991]:

$$h(y) = A(D_{50}) \times y^{2/3} \quad (8)$$

with $h(y)$[L] equal to the vertical distance below the highest point in the profile at a distance *y* from the landward start of the profile; and $A(D_{50})$[$L^{3/2}$] is a scale factor that varies with the sediment $D_{50}$ [L] size of the unconsolidated sediment. The Dean profile is fitted iteratively, as in SCAPE: if a talus and beach profile starting immediately seaward of the cliff cannot accommodate the required volume of sediment, then the whole beach profile is shifted one raster cell seaward, with the cell landward of the new start position (i.e. immediately seaward of the cliff cell) set to the same elevation as that of the cell on which the beach profile starts. This procedure is iterated, moving the beach profile further seaward each time, until the beach

profile can accommodate the required volume of talus. As the simulation progresses, the cliff raster cell is subjected to more wave energy and so the length of $L_1$ increases ($L_1 \rightarrow L_2$). When $L_2$ reaches a user-specified value $L_{max}$, cliff collapse occurs (Figure 9b). A volume of sand- and/or coarse-sized sediment with depth $d_2$ (i.e. from the base of the cliff notch to the elevation of the top layer of sediment on the cell) is removed from the cell, and deposited on the seaward cells as unconsolidated talus/beach. The simulation continues with a new notch being incised into the cliff cell (Figure 9c). Further collapses may occur as $L_3$ is extended. When the notch is eventually incised to the point that no further cliff collapse is possible on this cell (i.e. the total of all notch incision on this cliff cell equals the length of the cell side) then the cell is no longer flagged as a cliff cell. At the next iteration, the coastline-tracing procedure will treat this cell as a sea cell (Figure 9d). As the platform becomes wider, the energy reaching the cliff toe decreases and this reduces the rate of notch incision.

### 3.2.5 Alongshore sediment transport

Next, the alongshore unconsolidated sediment transport budget between all sediment-sharing polygons along the coast object is calculated. There are three stages to this. First, potential erosion/accretion (i.e. only transport-limited; not considering the availability of sediment) for each polygon is quantified using bulk alongshore sediment flux equations, and the direction of unconsolidated sediment movement (up-coast or down-coast) between adjacent polygons is determined. This results in a net potential unconsolidated sediment budget for each polygon. In the next stage, we consider only those polygons experiencing potential net erosion, according to this sediment budget. For these polygons, the availability of unconsolidated sediment in each of the three sediment size classes is quantified: this enables us to construct a between-polygon budget for actual (i.e. supply-limited) unconsolidated sediment movement, for the coarse- and sand- sized sediment classes only (eroded fine unconsolidated sediment is, in the current version of framework, simply stored globally). Finally, actual erosion or deposition is applied for the cells within each polygon in accordance with this supply-limited budget. These three stages are described in more detail below.

The conservation equation for beach sediment expressed in terms of local coordinates states that the change in position of the shoreline ($d\eta$), perpendicular to the local shoreline orientation ($s$) through time ($t$) is a function of the divergence of alongshore sediment flux ($Q_{ls}$).

$$\frac{d\eta}{dt} = f\left(\frac{dQ_{ls}}{ds}\right) \tag{9}$$

Typically, in bulk alongshore transport laws, flux depends on the height ($H_b$) and angle ($\alpha_b$) of breaking waves, e.g. CERC equation (Eq. 10) and *Kamphuis* [1985] equation (Eq. 11) (see comparison by *van Rijn* [2002]):

$$Q_{ls} = K_{ls} H_b^{5/2} \sin 2\alpha_b \tag{10}$$

$$Q_{ls} = 2.33\left(T_p\right)^{1.5} (\tan\beta)^{0.75} (D_{50})^{-0.25} (H_b)^2 [\sin(2\alpha_b)\,]^{0.6} \tag{11}$$

Where $K_{ls}$ [$L^{0.5}$ $T^{-1}$] is a transport coefficient $T_p$ is peak wave period; $tan\beta$ is beach slope, defined as the ratio of the water depth at the breaker line and the distance from the still water beach line to the breaker line; and $D_{50}$ is the median particle size in surf zone (m). Both the CERC and Kamphius equations estimate the potential immersed weight rate ($Q_{ls}$ [M $T^{-1}$]) for all active zone. In CoastalME, this is converted to bulk sediment transport rate per meter width ($Q_v$ [$L^3T^{-1}$]) by:

$$Q_v = Q_{ls}/\big((1-p)(\rho_s - \rho)g\big) \qquad (12)$$

where $p$ is the sediment porosity (~0.4), $\rho$ is the sediment density (~2650 [$ML^{-3}$] assuming quartz sand), $\rho_s$ is the water density (~1030 [$ML^{-3}$] for sea water) and $g$ is the acceleration due to gravity (~9.81 [$LT^{-2}$]).

In order to resolve Eq. (9), CoastalME next calculates the alongshore sediment fluxes in and out of each sediment sharing polygon. To calculate potential sediment transport on each polygon, average wave height and wave angle at breaking along each polygon's coastline segment- and the average beach slope are determined. Wave angle at breaking ($\alpha_b$) at each point along the coastline is calculated as the angle between the shoreline orientation (i.e. local, up-coast or down-coast) and the orientation of the wave crest. At each point along the coastline, the local coastline orientation is the angle (measured from the azimuth) of the straight line linking the two adjacent coastline points (i.e. the cell points up-coast and down-coast from the point of interest). Following *Ashton and Murray* [2006], as the local shoreline orientation is used to estimate wave angle at breaking for low-angle waves, and an upwind (i.e. shoreline orientation at next down-drift coast point) shoreline orientation is applied for high-angle waves. For coastal points where the changing shoreline orientation resulted in concomitant change in approaching wave angle at breaking from high ($\alpha_b < -45°$; $\alpha_b > 45°$) to low ($-45° < \alpha_b < 45°$) or vice versa, the flux maximizing angle $\alpha_b = \pm 45°$ is used. An average wave angle at breaking is calculated for each polygon as the average of all coast point wave angle at breaking. Average beach slope (as used in Eq. 11) is calculated as the ratio between the average depth of breaking and the average distance of breaking for all raster cells within each polygon.

Transport-limited conditions are assumed in COVE, such that there is always sufficient beach material available for transport. However, in CoastalME the actual alongshore sediment transport can be smaller than the potential bulk alongshore sediment transport (supply-limited conditions). The amount of unconsolidated sediment available on each polygon is defined by the sediment volume between an assumed equilibrium beach profile and the top elevation of the consolidated shore platform. In CoastalME the beach profile is assumed to have a user-defined equilibrium profile. The beach equilibrium profile currently assumed in CoastalME is the Dean profile (Eq. 8) [*Dean*, 1991]. By allowing the shore platform to adopt any slope, we do not need to use the analytical expressions used in COVE to calculate the shoreline changes as a function of volume changes, but instead use an iterative numerical scheme.

For each polygon with net potential erosion of unconsolidated sediment, CoastalME calculate the actual sediment flux by iteratively fitting Dean profiles in a landward direction until the volume of change from the start and end of the timestep equals the net potential erosion, or the profile becomes entirely consolidated. This is done by traversing the polygon's

coastline cells and fitting an equilibrium beach profile that is parallel to one polygon boundary (which is itself a coastline-normal profile). At each point along the profile, if the elevation of existing unconsolidated sediment is greater than the elevation of the assumed equilibrium profile, then some unconsolidated sediment is removed so that the elevation at that point becomes that of the assumed equilibrium profile. Sediment which is removed then becomes available for deposition elsewhere. But if, at that point, the elevation of the existing unconsolidated sediment is below that of the assumed equilibrium profile, then sediment is taken from the available sediment and deposited so that the elevation at that point becomes that of the assumed equilibrium profile. This is repeated for every point on the profile. If the available unconsolidated sediment from the whole profile is smaller than the target for potential erosion per profile for this polygon, the equilibrium profile is moved one cell landward iteratively until the available unconsolidated sediment equals the target for potential erosion, or all unconsolidated sediment at that coastline point is removed, or the cell is outside the grid. The target potential erosion per profile is obtained as the ratio between the polygon's previously calculated potential erosion sediment flux and the length of the coastline segment (units are $m^3$ sediment per $m$ of coastline per unit of time). This is repeated down-coast for every coastline point, and repeated traversing the coastline up-coast to ensure that no cells are missed. At the end of these iterative loops, the available unconsolidated sediment for this polygon is either equal to the potential sediment flux (if enough unconsolidated sediment is available on this polygon) or smaller (if constrained by the availability of unconsolidated sediment on this polygon). If a polygon has more than one adjacent polygon in the direction of sediment movement, then the fraction of total sediment volume exported to each of these adjacent polygons is assumed to be proportional to the shared length of boundary between these polygons, as in COVE.

A budget for actual unconsolidated sediment movement between each polygon may now be drawn up. For those polygons with net loss of unconsolidated sediment, the active layer availability equation (Eq. 7) is applied for each sediment fraction. This gives us the actual (supply-constrained) volumes of sand- and coarse-sized sediment lost from those polygons, and the net gain of unconsolidated sediment in adjacent polygons. At present, CoastalME just tracks the actual volume of eroded fine sediment: this is assumed to go into suspension, but in future developments we can incorporate transport rules for suspended material to make it available in estuarine settings. Actual elevation change (erosion or deposition) for unconsolidated sediment on each raster cell within each polygon is iteratively calculated as described previously, by fitting beach profiles: we search down-coast along the coastline of each polygon and fit beach profiles, iterating inland (for erosion) or seaward (for deposition) until each polygon's target is met; if it is not met then we traverse the coastline in the up-coast direction in case any cells have been omitted.

At the end of each time step, the framework outputs (if desired) spatial patterns as GIS raster or vector layers. It also outputs total sediment gains and losses for this time step. Finally, the updated raster grids (elevation plus stratigraphy) becomes the initial raster grids for the next time step. This loop is repeated until the end of the simulation is reached.

## 4 Examples of CoastalME composition outputs

On the previous section, we have shown how different models (CSHORE, COVE and SCAPE) have been integrated within CoastalME. Here we demonstrate the composition for different setup conditions. Validation of the composition will be treated in a separate, dedicated future study. With the current exercise, we aim to illustrate the emergent behaviours that the integrated framework can produce that is beyond the capability of the component models alone. The input files for each run can be downloaded from the project website (see code availability).

The initial conditions and main attributes used for the study cases presented below have some commonalities. The initial DEM is made of 1 m regular square cells with an extent of 1000 x 500 m and has an average slope in a seaward direction of 10.25°. SWL and wave forcing are assumed constant for the whole simulation (i.e. no sea level rise, tides neither storm/non-storm waves). The SCAPE rock strength calibration variable R is assumed to be the value used by *Walkden and Hall* [2011] for the soft-cliff coast of North Norfolk (East UK) and equal to $2\times10^6$ $m^{9/4}s^{3/2}$. The scaling factor for the alongshore sediment transport for the CERC equation is assumed as $K_{ls}$= 0.4 $m^{1/2}$ $s^{-1}$, sediment porosity is 0.4, and sand sediment density of 2650 $kg/m^3$. The DEM boundaries are assumed to be open boundaries where sediment is allowed to exit the domain but none enters. CSHORE is used as a wave propagation module (i.e. energy dissipation due to bottom friction and wave breaking is considered non-negligible). Run time of simulations will vary with the time step and the frequency at which outputs are written. For the test cases shown below (daily time step is used) one year is simulated in 180 seconds (i.e. run time is O(~$10^5$) faster than reality).

### 4.1. Role of sediment fraction composition on coastal change

We show how the integrated model, starting with the same DEM and forced by the same deep water waves and SWL but with different stratigraphy data, results in a different DEM evolution (Figure 10).

The initial DEM (Figure 10a) is made of two different sediment sizes compositions: (1) all DEM sediment is consolidated fine material (i.e. when eroded is lost in suspension), (2) 80% is consolidated fine and remaining 20% is consolidated sand. Wave forcing is constant with offshore significant wave height of 2 m, 10 seconds wave period and 225° wave direction. The differences on coastal response of these two initially-identical-topographies is clear after one simulated year (Figure 10a). In both cases the initial topography is eroded but to different degrees. The shoreline for the DEM made of fine sediments, has retreated as much as -40m in places, and a wide sub-horizontal shore platform and vertical cliff has been created. The shoreline for the DEM with a small percentage of sand has advanced seaward an average of 5 m. The reason why the shoreline shows different behaviours is better understood by looking at the shoreface elevation profiles at the start and at the end of the simulation (Figure 10b). For the case where only fine material is available, and SWL is constant, no beach is

created and the platform erosion rates drops asymptotically as the platform is widening and dissipating more wave energy through wave breaking. For the case of the DEM with a 20% sand fraction, erosion at the coast provides sand to allow a thin fronting beach to form, reducing the erosive potential of the waves by protecting the consolidated platform beneath it. After a few time steps, once the beach is formed, the eroded sand is then only lost at the boundaries of the domain, driven by the alongshore sediment transport gradient. In the case of an initial fine-sized DEM, the only process able to reduce coastal erosion is platform widening, while in the case of the mixed fine and sand cliff a new process (i.e. beach platform protection) emerges as soon as the beach thickness is sufficient to provide protection against the breaking waves. Beach width further controls the amount of sediment lost from the domain by controlling the gradient of the alongshore sediment transport.

### 4.2 Effect of a weak zone in a continuous line of coastal defences

We show how a weak segment of a long continuous line of defence results in the formation of a bay and cliff on an initially rectilinear and gently sloping coastal landscape (Figure 11). A horizontal breakwater protects all but one segment O(~100 m) of the coast at about the centre of the domain (Figure 11a). Wave forcing is constant with offshore significant wave height of 2 m, 10 seconds wave period and 270° wave direction. The SWL is constant and equal to 60 m above the basement. All DEM sediment is consolidated fine material (i.e. when eroded is lost in suspension).

After three model years of simulation, results show how an initially straight coastline develops a small cliffed bay at the undefended segment of the coastline (Figure 11b). The breakwater is not at the coastline at the start of the simulation. After about 90 days of simulation the consolidated platform at the seaward side of the breakwater is eroded and the shoreline retreats. Once the shoreline reaches the breakwater, no more landward erosion occurs on the protected coastline but erosion continues along the un-protected shoreline where defences are damaged. After a year, a small bay has emerged and evolves asymptotically towards a circular-shaped bay after three years of simulation. The resulting bay is bounded by a vertical cliff (Figure 11c). Similar embayments can be found in nature, for example along the south coastline of the UK (Figure 11d) and many other places worldwide.

### 4.3 Interruption of alongshore sediment transport by a groin

We show how a groin interrupts the alongshore sediment transport creates accumulation and erosion patterns as typically observed in nature (Figure 12). A perpendicular groin of 84 m length (from the shoreline) is located at about the centre of the simulated domain. Wave forcing is constant with offshore significant wave height of 2 m, 10 seconds wave period and 225° wave direction. The SWL is constant at an elevation equal to 85 m above the basement. All DEM sediment is consolidated sand material (i.e. transport-limited condition). Alongshore sediment transport for this simulation is from north to south.

After one year of simulation the initially straight coastline has advanced and prograded at different sections along the coast (Figure 12b). At the updrift side of the groin, sediment is accumulated at the beach but also along the groin exposed face and bypasses the groin tip to be deposited on the downdrift side. No eroding cliff is formed within the shadow zone. As typically observed, erosion occurs at the downdrift side of the groin where the shadow zone intersects the shoreline, due to limited sediment supply resulting in a negative flux gradient.

# 5 Discussion

## 5.1 CoastalME as a modelling framework of large scale coastal behaviour

Modularity is a fundamental requirement of the design of CoastalME. As discussed previously, the CoastalME framework captures as software, of the 'essential characteristics' of component models. Therefore, if a user wishes to replace one component model with another, it must be made relatively easy for one part of the framework to be 'exchanged' with an equivalent software component that provides the same functionality. In this work, we have demonstrated the integrative capacity of this novel framework by implementing several component models (Table 3).

To achieve this kind of plug-in modularity, CoastalME adopts the object-oriented architecture design and programming paradigm [e.g. *Rumbaugh et al.*, 1991]. Conceptually, the modelling framework comprises software objects, which are instances of software classes (Figure 13). The software classes which comprise CoastalME are themselves categorised. They may represent geometrical constructs such as a point, a line or a raster cell; or real-world objects such as a coastline, a cliff or an intervention (these latter being drawn from the ontology shown in Table 2). The inputs and outputs of each software object are clearly specified (see code availability): this in theory enables one software object to be replaced with another, if both offer identical inputs and outputs. Similarly, the framework provides base software elements for the implementation of new model components.

There are, nonetheless, practical limits to this modularity. Whilst the most straightforward modularity would just involve re-implementing an existing software object with an equivalent model that provides slightly different functionality, this replacement might however also require additional inputs. This could be the case if, for example, the user wished to try a different equation for alongshore bulk sediment transport. A more ambitious re-implementation of parts of the framework would certainly require extra inputs: this would be the case if replacing the current alongshore uniform wave routing routines with a more physically-based approach. Replacing or supplementing other aspects of the CoastalME framework would require considerable re-design. Using an approach other than the current polygon-based scheme for routing unconsolidated sediment would, for example, be challenging, but the basic geometric objects can provide the building blocks for implementation of alternative models.

The model framework (currently about 17000 lines of C++) uses only standard C++ libraries to maximize portability, with two exceptions: the Geospatial Data Abstraction Library (GDAL, 2017) which is used to read and write the GIS outputs, and to interpolate wave attributes from coastline-normal profiles to grid cells and a Fortran library that is call if the user selects CSHORE as the wave transformation approach. Thanks to the functionality of GDAL, CoastalME is highly flexible regarding the raster input formats which it can read, and the raster and vector output formats which it can output. The user-preferred raster input/output format is defined, among others, in the configuration parameter file (see code availability).

### 5.2 Behaviour of the CoastalME composition: COVE-CSHORE-SCAPE

The CoastalME composition presented in this work has several capabilities than make the integrated model more appealing than using the individual models in isolation (Table 4). The most obvious additional capabilities relative to COVE as a stand-alone model is the ability to represent cliff and shore platform erosion and beach interaction (i.e. COVE is limited to unconsolidated sediment alone) and the new additional capability relative to SCAPE alone is the ability to reproduce highly irregular coastlines. Less obvious, but equally important, is the added capability, for both SCAPE and COVE, of capturing the effect of the regional bathymetry on the local alongshore sediment transport and the energy dissipation due to wave breaking and bottom friction (i.e. if CSHORE is used as wave propagation module). As with other one-line models (e.g. Hanson and Kraus, 2011), the offshore contour orientation in SCAPE and COVE upon which the incoming waves are refracted is assumed to be parallel to the shoreline orientation. This assumption ensures that the incident waves are realistic while preserving feedback between shoreline change and the wave transformation. However, the assumption has a limitation: an open coast without structures or sources and sinks of sediment will evolve to a straight line if a standard shoreline response model is run for a sufficiently long time. In the integrated CoastalME model, waves are propagated upon the full DEM and therefore the local gradient of the alongshore sediment transport is a combination of the local orientation of the shoreline and the regional orientation of the bathymetry (i.e. regional bathymetry controls wave propagation). Figure 10 illustrates the effect of this regional bathymetry influence on the two simulated cases. For the case of DEM being made of all fine consolidated sediment, the shoreline retreats following the regional bay-shaped bathymetry. The sediment sharing polygons at the end of the simulation are similar to the ones at the beginning, but translated landward. For the case of mixed consolidated fine and sand DEM, the shoreline also follows closely the regional bay shaped bathymetry since most of the beach sediment volume is at the shoreline. In this last case, the sediment sharing polygons at the end of the simulation have not only being translated landward but also have a more intricate sediment sharing pattern to the former ones.

### 5.1 Dynamically linked raster and vector objects to represent coastal change

CoastalME's representation of space, and of the changes occurring within its spatial domain, involves both raster (i.e. grid) and vector (i.e. line) representations of spatial objects. This is commonplace in modern GIS packages. What is relatively

unusual, however, is that in the CoastalME framework data is routinely and regularly transformed between these two representations during each time step of a simulation. This may appear both perverse and computationally inefficient: however, there are advantages which will be discussed below.

A coast is an approximately linear boundary between sea and land: hence (and unsurprisingly) coastal modelling has a strong historical emphasis on linear – i.e. vector – models (see the discussion of LSCB models). It was clear from the outset that CoastalME would build upon this tradition and so would use 2D vector representations of coastal features.

However, a raster grid – comprised of multiple cells, usually square or rectangular – is a widely-used alternative approach to representing 2D space. Raster grids have several attractive features when used for the acquisition, storage, and manipulation of spatial data [e.g. *Densmore et al.*, 1998]. Data such as topography are readily available in grid form and linkage with other environmental models is facilitated, since such models often output their results as raster grids. Also, Cellular Automaton (CA) models operate upon regular grids and have taught us much regarding the spatial patterns produced by emergent behaviour [e.g. *Dearing et al.*, 2006; *Favis-Mortlock*, 2013b; *Murray et al.*, 2014]. Thus, at an early stage of development it was recognized that using raster grids for data input, storage and output, would provide a consistent framework for handling sediment exchange (and hence sediment mass balance), whilst a variety of raster and vector representations could be used to describe morphological change.

A raster grid also has several disadvantages. The first is the creation of axially-aligned spatial artefacts: it is not trivial to ensure that cell-to-cell movement is uninfluenced by the alignment of the grid's axes. To achieve this invariably involves some computational expense. There is also the problem of spatial precision and computational needs. The cell is the smallest spatial unit of the grid, so small spatial features can only be adequately captured by using small grid cells. Similar reasoning applies when there is a need to represent cell-to-cell flows that are fast-moving relative to cell size: for explicit formulations, the Courant-Friedrichs-Lewy Condition requires that the time step must be kept small enough for information to have enough time to propagate through the discretized space [*Weisstein*, 2016], yet this can dramatically increase computation time. A third consideration regarding cell size results from the tendency of the dominant geomorphological process to change with change in spatio-temporal focus [*Schumm and Lichty*, 1965]. But as raster cells shrink or grid sizes grow, computational requirements increase non-linearly. If the majority of grid cells are involved in computation during most of the simulation, the increase in computational requirements may be roughly the square of the ratio of decrease in cell side, or even worse [*Favis-Mortlock*, 2013b].

Yet coastal modelling – with its strongly line-oriented focus – does not require an egalitarian treatment of grid cells. The computational focus need only be on the coastal zone, i.e. on a subset of grid cells; with much less happening, computationally-wise, on the remainder of the grid. So we reasoned that despite our need for small cells (since we would be

dealing with small features, and sometimes with fast-moving fluxes) and hence a large grid, only relatively few cells within that grid – those on or near the coastline – would require computationally expensive treatment. This was reassuring, but there is of course a computational overhead associated with conversion of spatial features between vector and raster representations. By contrast, the model's treatment of simulated time is conventional. There is a fixed time step that can, in principle as for any implicit method, be of any duration. In practice, however, there are constraints on time step duration due to the amount of sediment that can be eroded or deposited in a single time step without unrealistically de-coupling the morphology change and assumed hydrodynamic forcing during a given time step [*Ranasinghe et al.*, 2011].

In summary, CoastalME's hybrid raster-vector structure involves a trade-off between increased complexity (due to the need to transform between raster and vector representations) and parsimonious spatial structure (because the majority of computation involves only cells on or near the coast).

## 6 Conclusions

Numerical modelling of complex coastlines requires consideration of interactions between multiple coastal landforms. Despite efforts to couple separate models (e.g. software wrappers such as OpenMI and CSDMS), there is a need to deal more directly with the semantics of the various entities being modelled. We have presented here a description of, and proof-of-concept results from, a flexible and innovative modelling framework (CoastalME) for integrated coastal morphodynamic modelling at decadal to centennial timescales and spatial scales of 10s to 100s km (mesoscales). To achieve this, CoastalME integrates the concept underlying each model as a set of dynamically-linked vector and raster objects.

The rationale underpinning CoastalME results from the observation that most of the existing simulation models for coastal morphodynamics at meso-scales conceptualize the real complex 3D topography of the coastal zone using simplified geometries. Accordingly, we have devised a spatial framework which is consistent with these simple geometries, and which permits the representation of these existing models in terms of behavioural rules which operate within this spatial framework. Thus, the DEM and stratigraphy is represented as a raster grid of regular cells, each of which holds some thickness of consolidated and unconsolidated sediment which is itself comprised from three size fractions (coarse, fine, sand). Vector-based spatial objects are created at each time step that represent features such as the coastline, profiles which are normal to that coastline, and polygonal coastal cells that are partially bounded by these normal profiles. Driven by external boundary conditions (waves, currents and sea level), coastal processes which mobilise sediment are simulated using these vector-based objects, and the resulting changes to the spatial distribution of sediment are then stored in the raster grid. Modelled topography therefore changes as each cell's store of sediment changes its thickness during a simulation, with

sediment being eroded in some cells and deposited in others, maintained in suspension or lost at the boundaries due to external boundary conditions (waves, currents and sea level changes). In addition to the set of blocks or raster objects, the authors suggested a minimum set of classes needed to reproduce a generic morphodynamic model. We suggest that a variety of existing coastal models, each of which represents a single landform element or a limited range of elements, contributing to coastal morphodynamics (e.g. estuary, salt marsh, dunes etc.) may be integrated within CoastalME's modelling framework. As a proof-of-concept example, we have integrated a one-line model for very irregular sediment-rich coastlines with a soft cliff and beach erosion model. We then verify that the integrated models behave as expected, for example by; (1) demonstrating that given the same initial topography and forced by the same external drivers, differing stratigraphic inputs produce different coastal morphologies; (2) showing how a weak segment on coastline of defence can evolve into an embayment and; (3) how a groin can partially block the alongshore sediment transport creating zones of accretion and erosion.

## 7 Code availability

The CoastalME is developed and maintained within the GitHub web-based repository hosting service. This repository allows users to download frozen versions of the model (version 1.0 at the time of writing) to keep their local copy up to date. The version 1 can be found in https://github.com/coastalme/coastalme. The folder structure at the github repository contains the input files used for the test cases shown in section 4. A dedicated wiki-site to CoastalME which includes the model documentation, user manual, test cases, software requirements, installation guide, related publications and reports and a note about the framework developers can be found in http://www.coastalme.org.uk/. This Wiki site includes a section on frequently asked question. Any question regarding CoastalME can be emailed to admin@coastalme.org.uk.

This code is also available from the iCOASST project modes dedicated web site at the Coastal Channel Observatory web site (http://www.channelcoast.org/iCOASST/introduction/). The user accessing the code through this route will be able to see how CoastalME framework related with other existing modelling approaches of decadal and longer coastal morphodynamic.

CoastalME is free software; you can redistribute it and/or modify it under the terms of the GNU General Public License as published by the Free Software Foundation; either version 3 of the License, or (at your option) any later version. This program is distributed in the hope that it will be useful, but WITHOUT ANY WARRANTY; without even the implied warranty of MERCHANTABILITY or FITNESS FOR A PARTICULAR PURPOSE. See the GNU General Public License for more details. The user receives a copy of the GNU General Public License along with this program; if not, write to the Free Software Foundation, Inc., 675 Mass Ave, Cambridge, MA 02139, USA.

*Acknowledgements*. This work was funded by the Natural Environment Research Council (NERC) as part of the Integrating COAstal Sediment SysTems (iCOASST) project (NE/J005541/1), with the Environment Agency as an embedded project stakeholder. Special thanks to Dr. Bradley Johnson from the USA Army Corp of Engineers for allowing us to include the executable of CSHORE as part of this composition. This paper is published with the permission of the Executive Director, BGS (NERC).

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

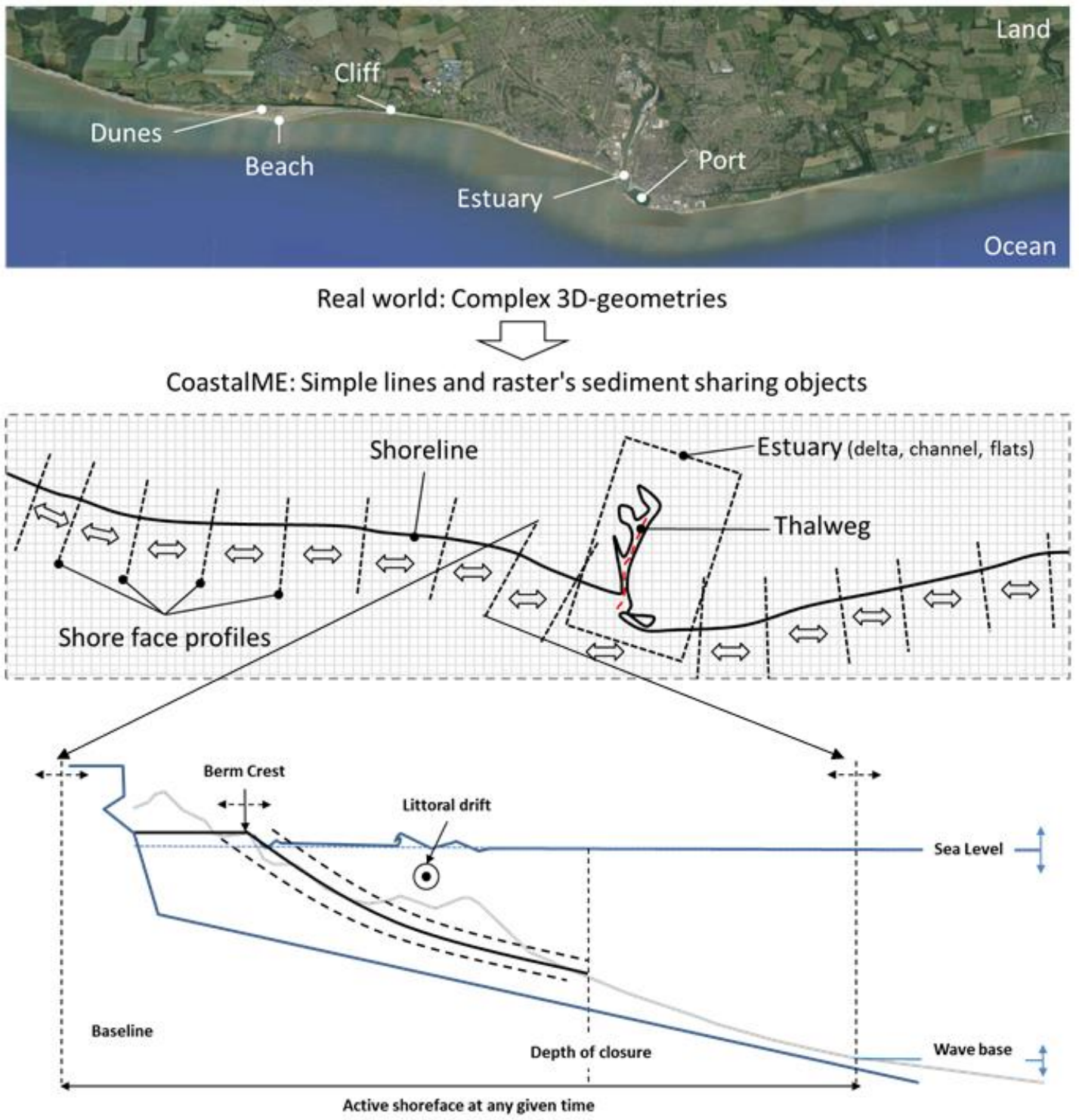

**Figure 1: Schematic diagram of the proposed modelling approach. Coastal morphology change is simulated as dynamically linked line and raster objects. The hierarchy of panels illustrate how a real coastal morphology (upper panel) is conceptualized as shoreline, shoreface profiles and estuary elements (middle panel). All elements can share sediment among them (double-headed arrow). The shoreface comprises both consolidated and non-consolidated material that forms the cliff, shore platform and beach respectively (bottom panel). At every time step the shoreline is delineated at the intersection of the Sea Level and the ground elevation. Shore face profiles are delineated perpendicular to the shoreline. The Sea Level and wave energy constrains the proportion of shoreface profiles that are morphologically active at each time step. Eroded sediment from the consolidated profile is added to the drift material to advance the shoreline or loss as suspended sediment. Gradients of the littoral drift further controls the advance and retreat of the beach profile and the amount of sediment shared with nearby sections of the shoreline.**

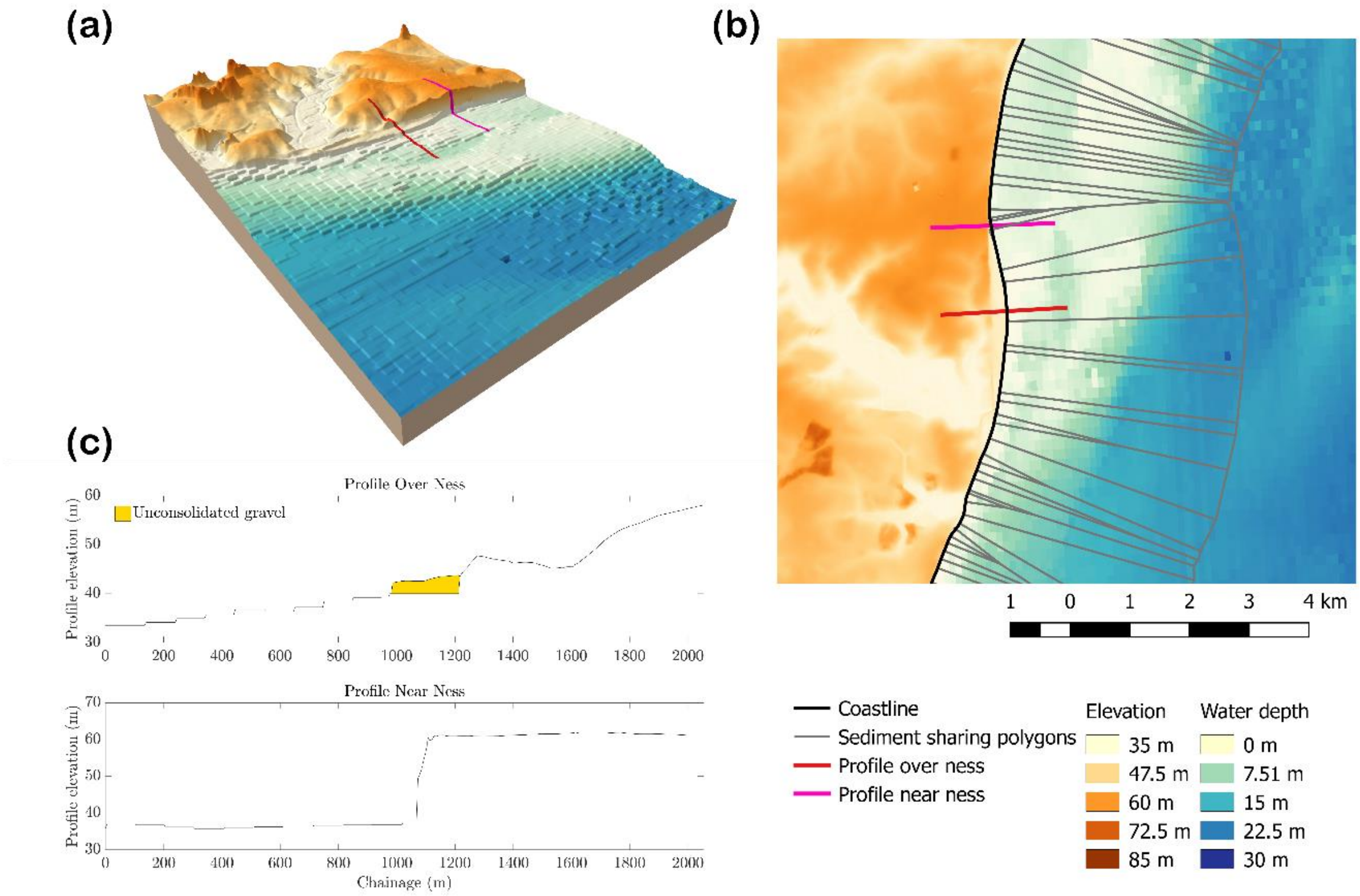

**Figure 2. In CoastalME, the shoreface is conceptualised as a set of sediment sharing cells interconnected by the alongshore sediment transport: (a) Detailed (5m x 5m) Digital Elevation Model of Benacre Ness in the East coast of UK, (b) sediment sharing polygons created by CoastalME, (c) example of a two profiles, showing a combination of consolidated and unconsolidated layer (profile over Ness) and purely consolidated (profile near Ness).**

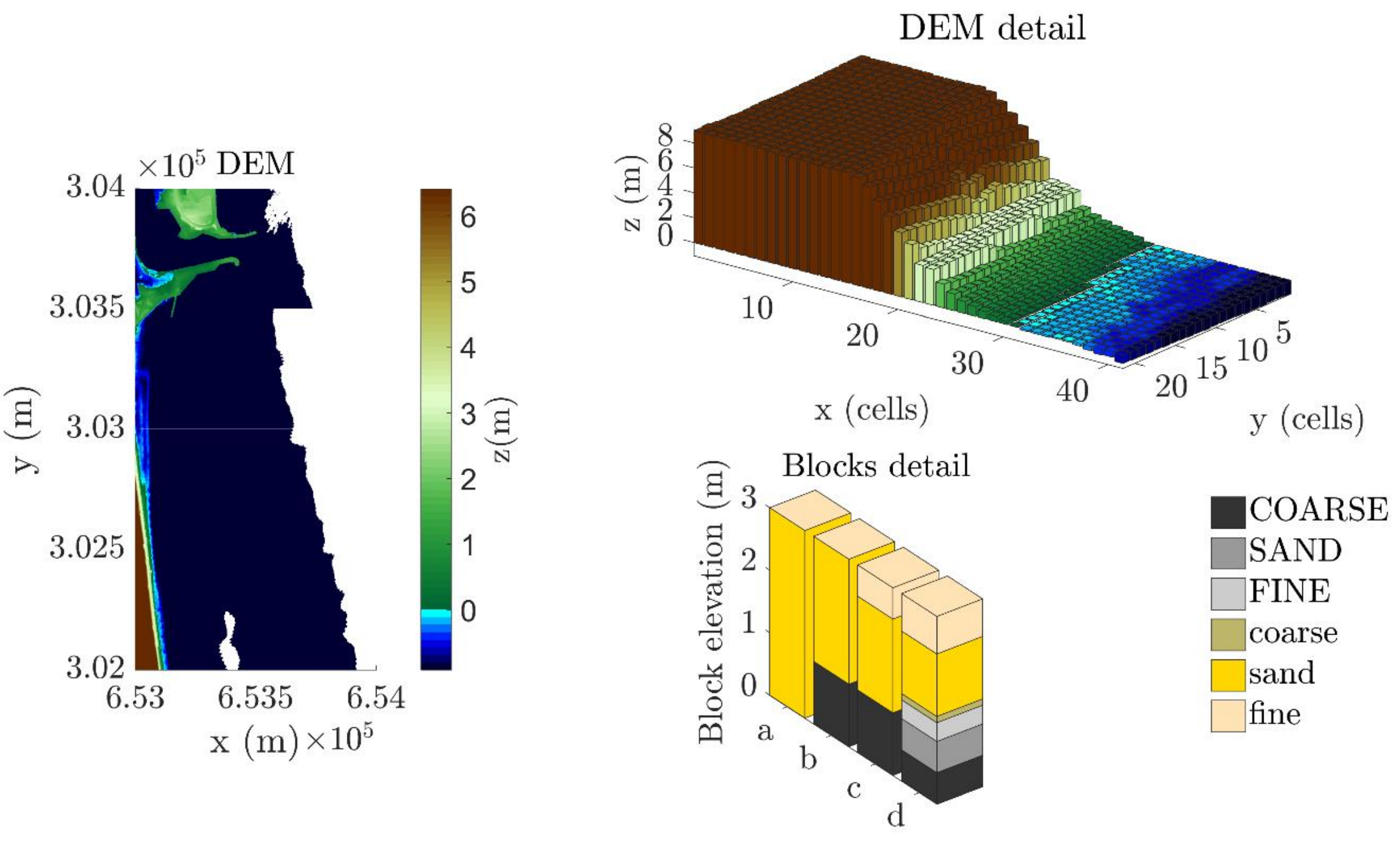

**Figure 3. Ground elevation is characterized as a set of regular square blocks. Each block has a global coordinate x, y, z. Left panel illustrates the DEM of Gorsleston-on-Sea (East coast of UK) provided by © Environment Agency copyright and/or database right 2015 with a raster resolution of 2m (DEM detail). Each block might be composed of six different sediment fractions (Blocks detail) made of coarse, sand and fine sediment sizes. Each sediment size fraction can be in a consolidated (capitalised) or unconsolidated state (lower case). Block types a, b, c and d illustrate blocks of same total elevation but with different sediment composition.**

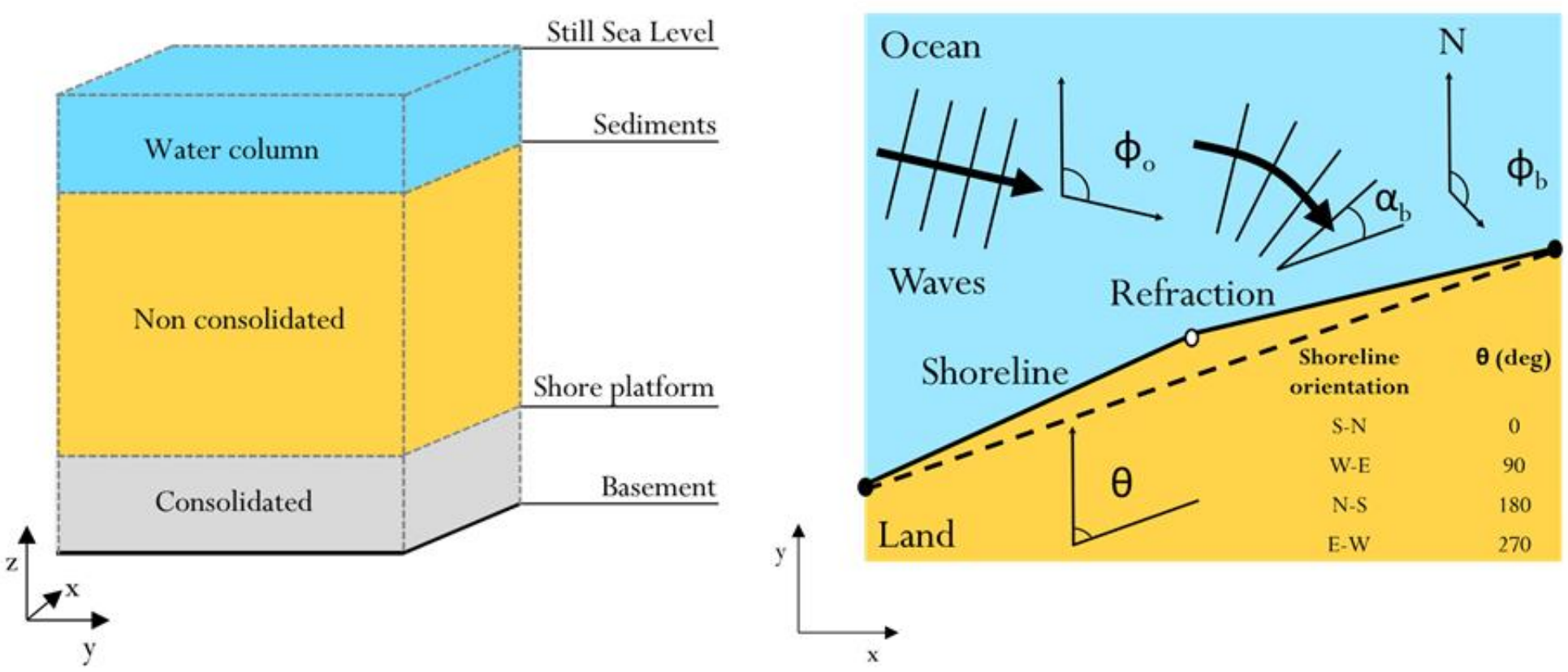

**Figure 4. Convention used in CoastalME for global coordinate system and wave direction. All layers (consolidated and unconsolidated) and sea elevations are referenced to a basement level (left panel). Shoreline orientation at each coastline point is defined as the angle relative to the azimuth clockwise which forms the straight line that connects the coastline points before and after (black dots) "this" (white dot) coastline point. Wave angle at breaking is obtained from the difference between the shoreline orientation and the wave front.**

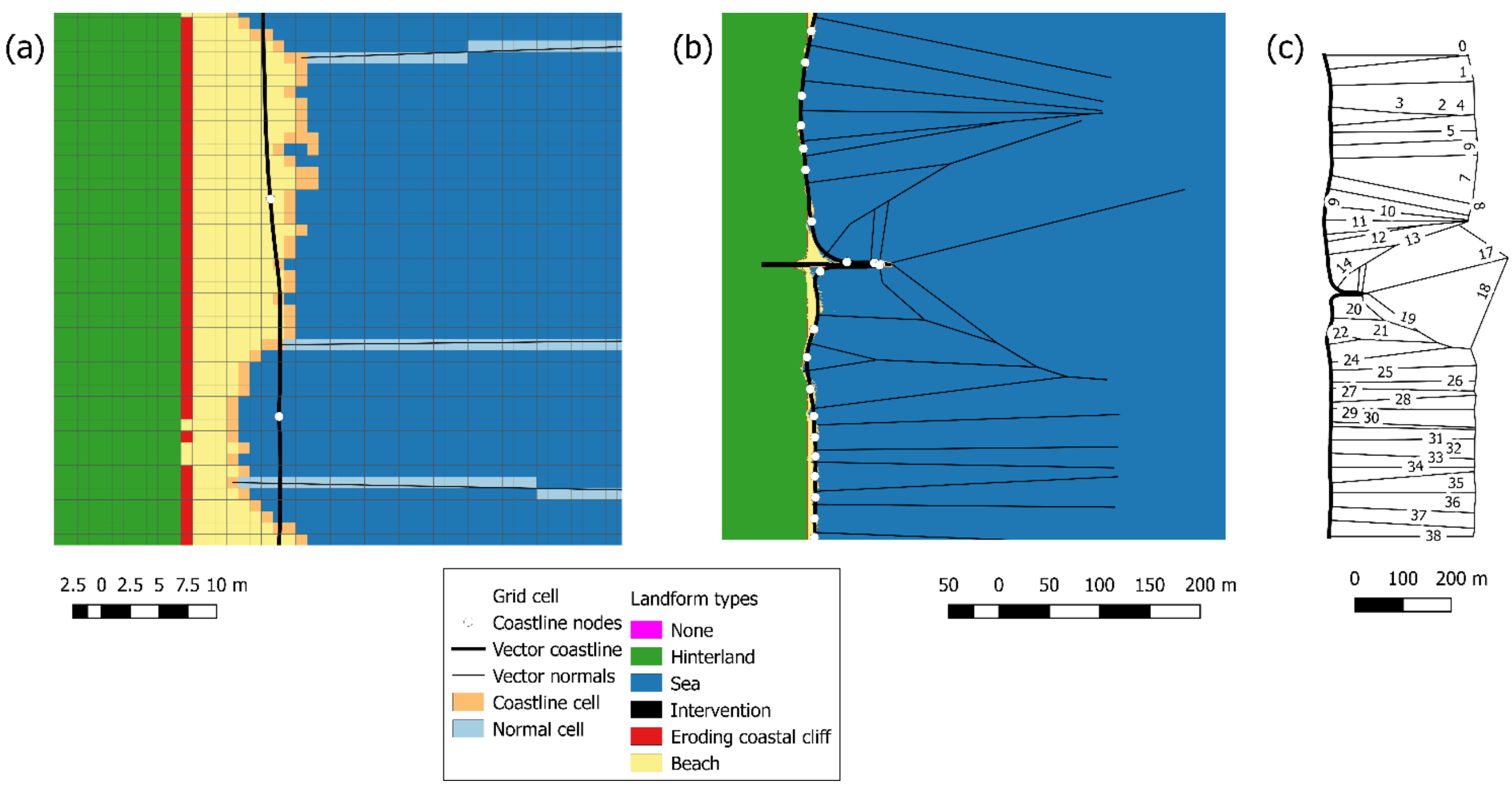

**Figure 5. Schematic diagram of CoastalMe landform classification mapping, and how raster and vector coastline are used to creat sediment sharing polygons. (a) Detail of raster coastal points and the smoothed vector coastline used to draw the coastline normal. Each cell of the grid is mapped as a landform type that will be later used to apply different behavioural rules (i.e. eroding coastal cliff). (b) Coastline normal are projected seaward and merged if the intersect before the end of the user defined normal is reached. (c) Triangular and trapezoidal sediment sharing polygons are created using the merged coastline normal as polygon edges. Polygons at the boundary of the grid are created differently to ensure that all cells are within the grid.**

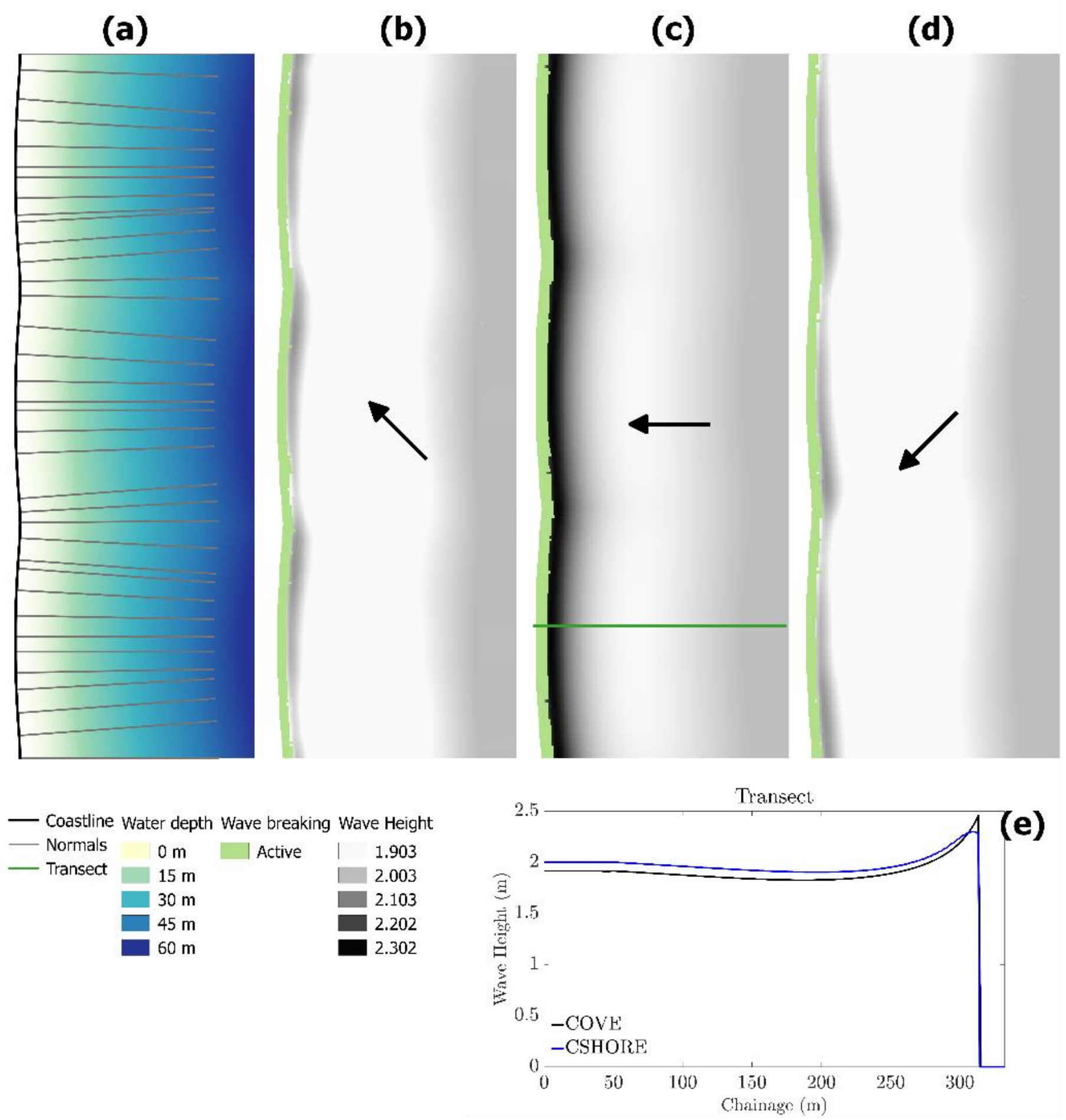

**Figure 6. Two wave propagation modules are integrated in CoastalME. Deep water waves are propagated along each coastline normal (a) and then the results interpolated to all grid cells. The wave height distribution for incoming waves at 315°, 270° and 225° (using the CSHORE module) are shown in panels b, c, d respectively. A comparison of the wave height distribution along the same transect using CSHORE and COVE approaches is shown in panel (e).**

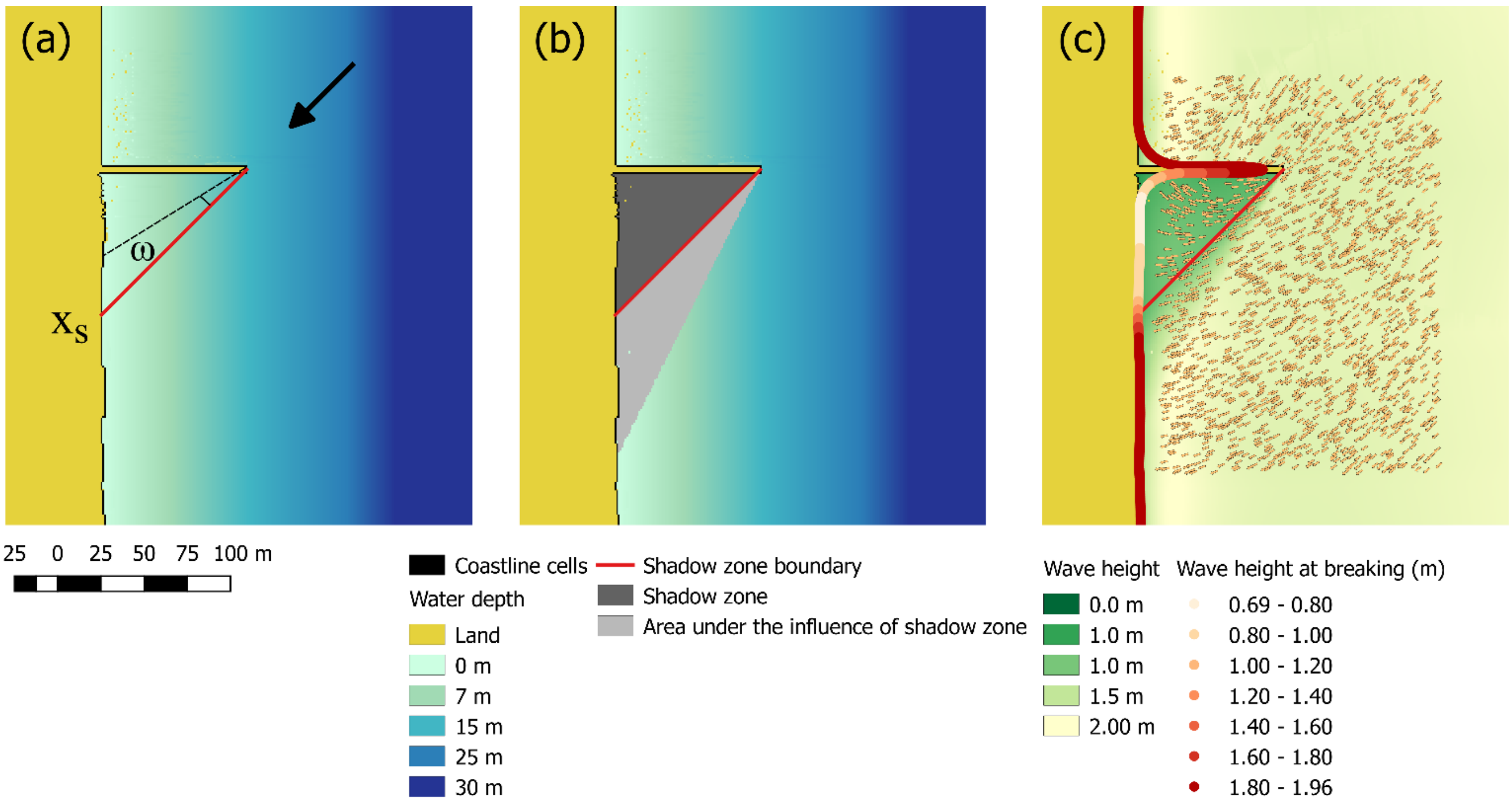

**Figure 7. Schematic diagram of a groin to illustrate shadowing and wave adjustment in the shadowed region. (a) The shadow zone is generated with respect to the offshore wave direction (black arrow). The angle within the shadow zone is defined by $\omega$. (b) The length of coast affected by rules for diffraction is twice the length of the shadow zone as shown by the sea cells marked as either within the shadow zone (dark grey) or on the area of influence of the shadow zone (light grey). (c) Adjustment of wave approach angle (arrows) by factor 1.5 times the angle within the shadow zone $\omega$ (equation (2)). The adjustment only proceeds up to $\omega$ = 90 since wave heights are zero beyond this value. Reduction in wave height due to wave crest spreading, which is defined by a sinusoidal function (equation (3)) with the wave height at the edge of the shadow zone assumed to be reduced by a factor of 0.5 with that factor increasing to 1 at $\omega$ = 90. Wave heights outside the shadow zone are also reduced to conserve wave energy following equation (4). Properties of wave at breaking are stored for into a vector coastline. Coloured dots in panel b shows wave height at breaking.**

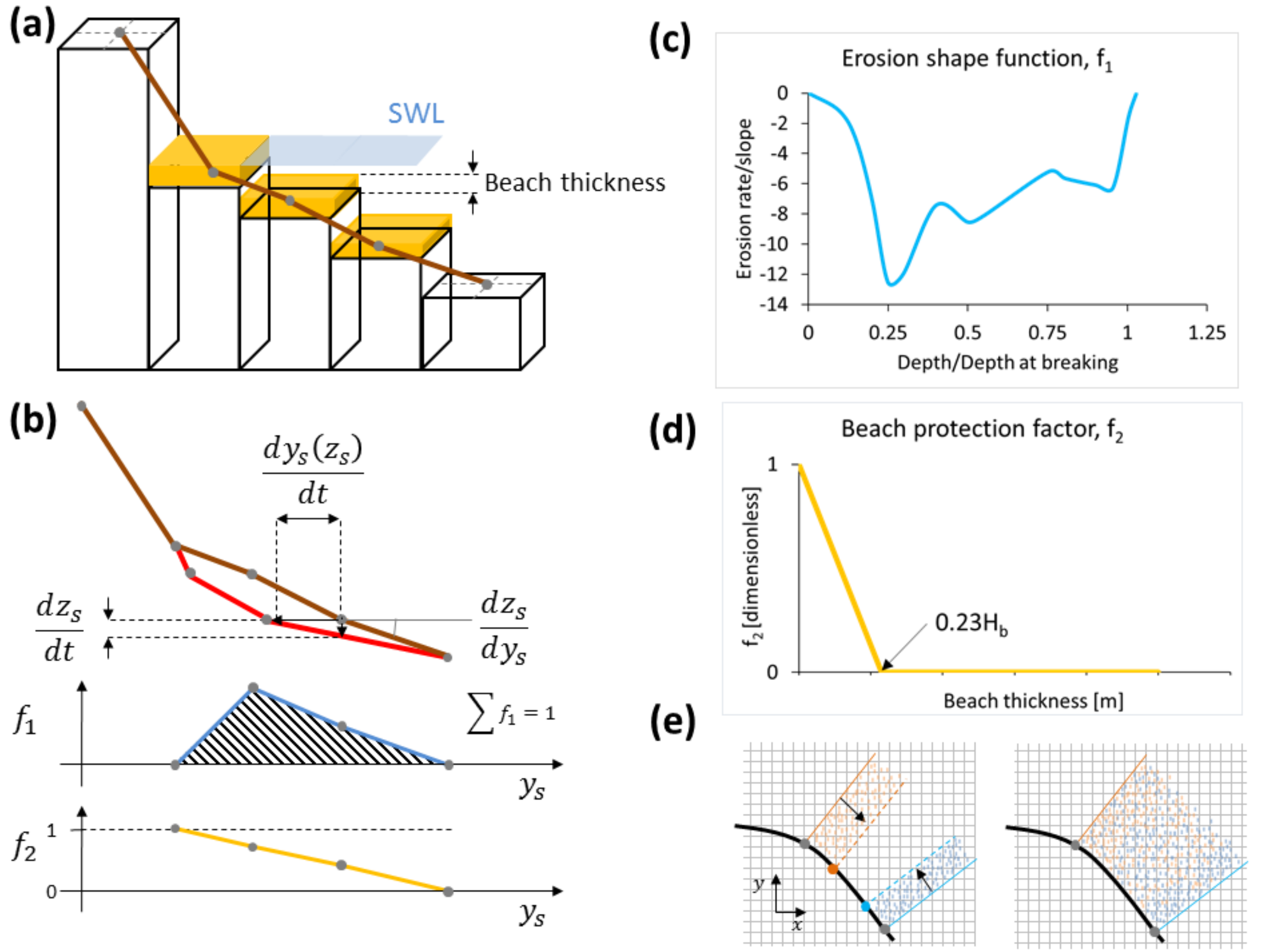

**Figure 8. Schematic diagram illustrating how the SCAPE concept of down-wearing of the consolidated shore platform is integrated in CoastalME. (a) CoastalME represent the profile as a set of vertical blocks of consolidated and unconsolidated material. The profile of the consolidated shore platform (brown line) is obtained by querying the top elevation of the consolidated material. Beach thickness on each block is represented as the thickness of any unconsolidated material on top of the shore platform. (b) The shape function used in SCAPE is queried as a look up table along each cell of a profile to estimate the erosion potential for a given wave height at breaking and water depth ($f_1$ in equation 5). (c) The beach thickness protection factor is also calculated for each cell. Shore platform is fully protected if thickness is larger than 0.23 times the wave height at breaking ($H_b$) and protection factor linearly increases to 1 as beach thickness decreases to zero ($f_2$ in equation 5). (d) The SCAPE's horizontal shore platform erosion component is converted to its vertical component using a trigonometric conversion (equation 6). The sum of all $f_1$ values along a coastline profile are equal to 1. (e) Shore platform erosion for each cell between coastline normal is calculated as explained above and by traversing the coastline cells (in both directions) using temporary profile parallel to the right and left boundaries. CoastalME checks that shore platform erosion is calculated for all cells within the surf zone.**

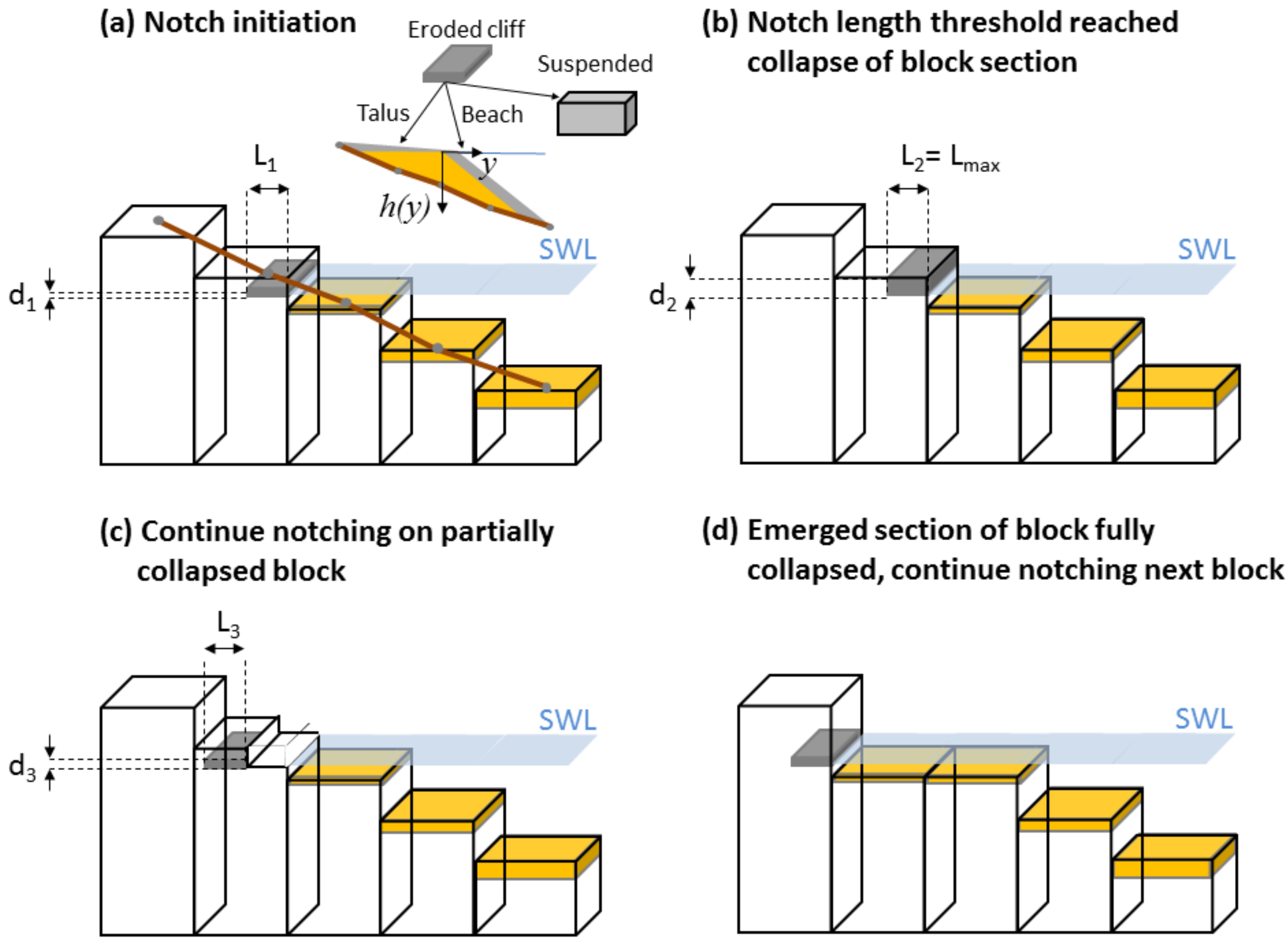

**Figure 9. Illustration how notch evolution (from SCAPE) is simulated in CoastalME. (a) Notch initiation represented as vertical blocks. Eroded sediment from the cliff is transferred to a talus/beach (coarse and sand fractions) and total suspended sediment (fine-sized sediment fraction). (b) Cliff collapses if the threshold cliff notch length is reached. (c) Notching continues on a partially collapsed block until the block is fully eroded (d) and notching may continue in the next-landward block.**

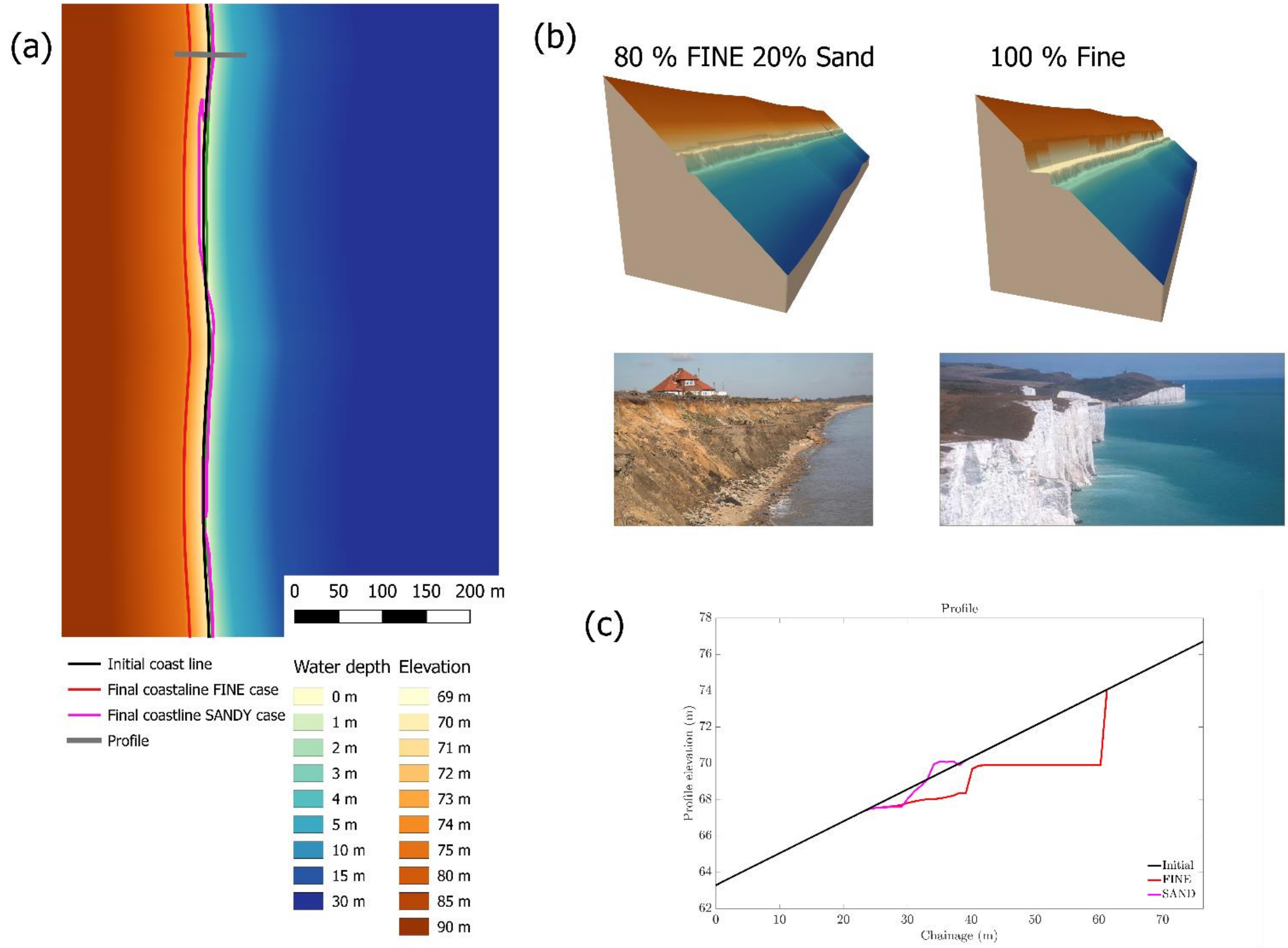

**Figure 10. Simulation results showing how sediment fraction composition affects the shoreline responses. (a) Both simulated cases start with the same cuspate shoreline and gently sloping DEMs but shoreline after one year (coloured lines) shows opposite responses (retreat/advance). (b) DEM containing only fine sediment is eroded creating a cliff fronted with a horizontal platform, like the chalk cliff of East Sussex (UK). Eroded sediment from the Sandy DEM forms a protective beach and no cliff. (c) Elevation profiles at a cross section reveals how the eroded sediment from the submerged platform has formed a protective beach and how a horizontal platform has been created.**

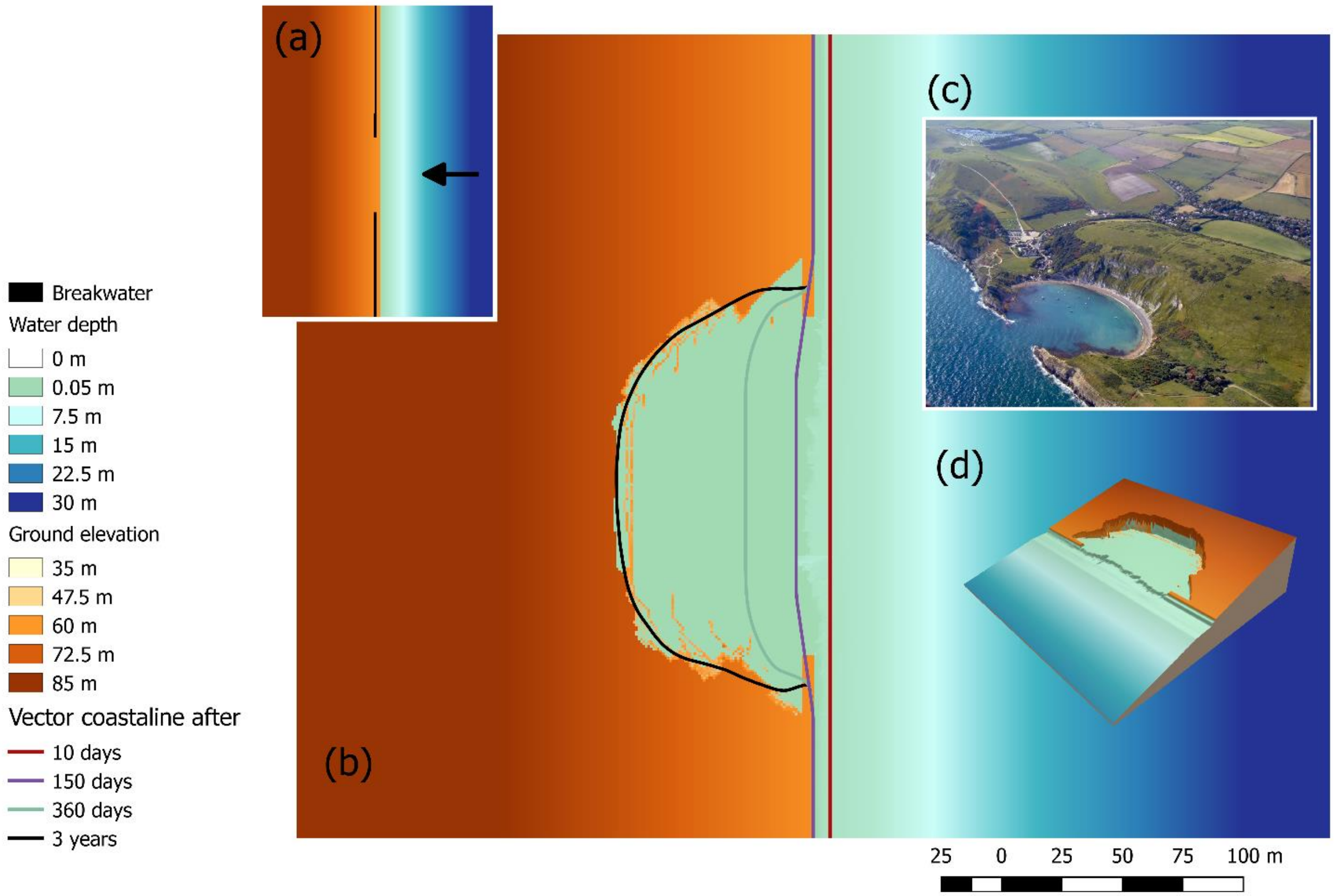

**Figure 11. Simulated embayment creation on an initially rectilinear coastline. (a) At the start of the simulation, all the coastline of a gently sloping topography is protected by a breakwater but a short segment in the centre that is un-protected. (b) Location of the vector coastline at different time steps and final topography after three years of simulation. (c) The resulting embayment is bounded by a cliff similar to the Lulworth Cove bay in the south of the UK.**

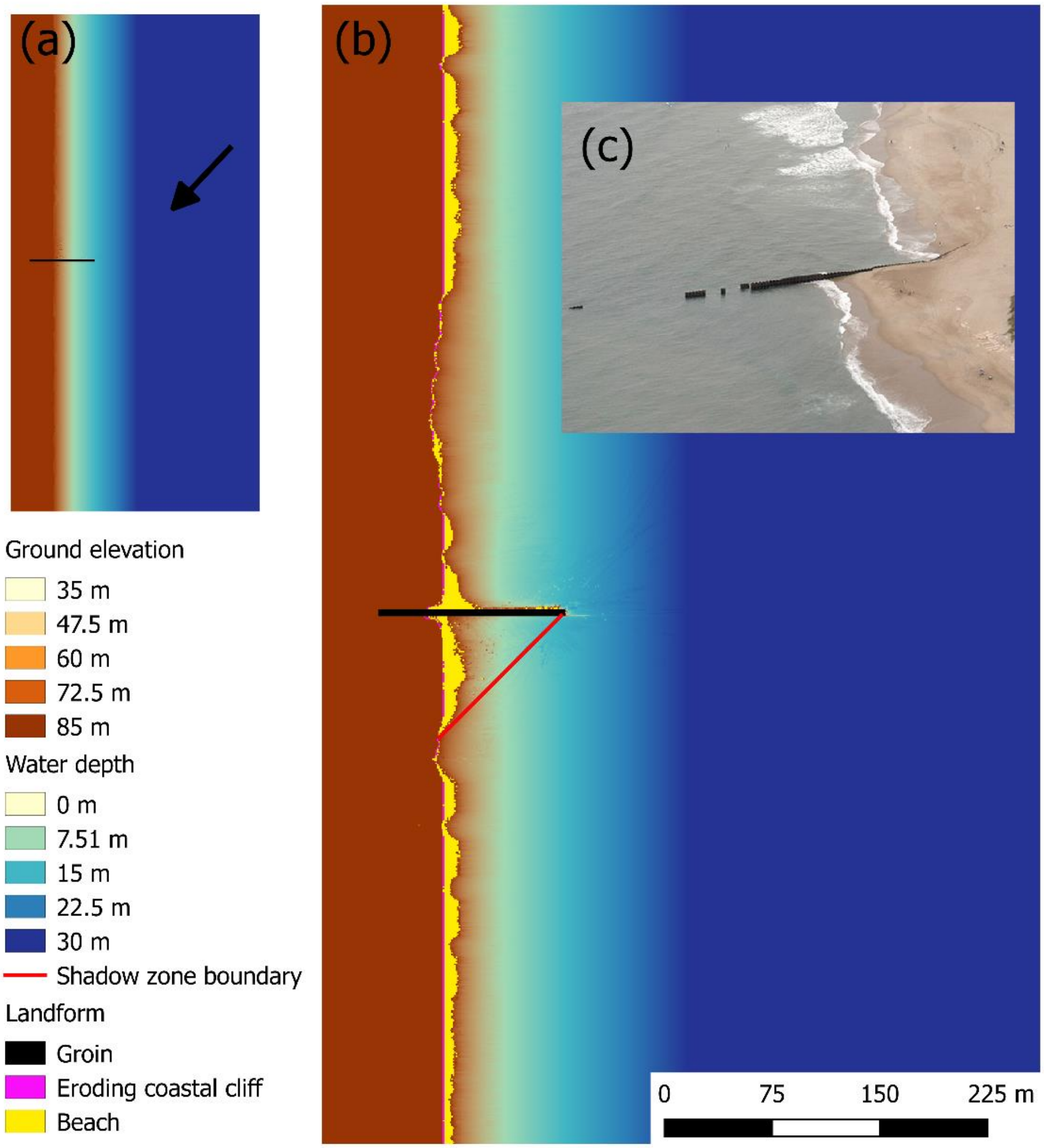

**Figure 12. Simulation results showing how a groin interrupt the alongshore sediment transport. (a) Simulation starts with a rectilinear and gently sloping DEM interrupted by a groin and waves coming at 225° (arrow). (b) After one year of simulation an eroding cliff and a beach of different widths is created. Sediment is accumulated at the up-drift side of the groin and eroded on the down drift side where the shadow zone intercepts the coastline. (c) Shows a groin with accumulated sand on the up-drift side and less sand on the downdrift side.**

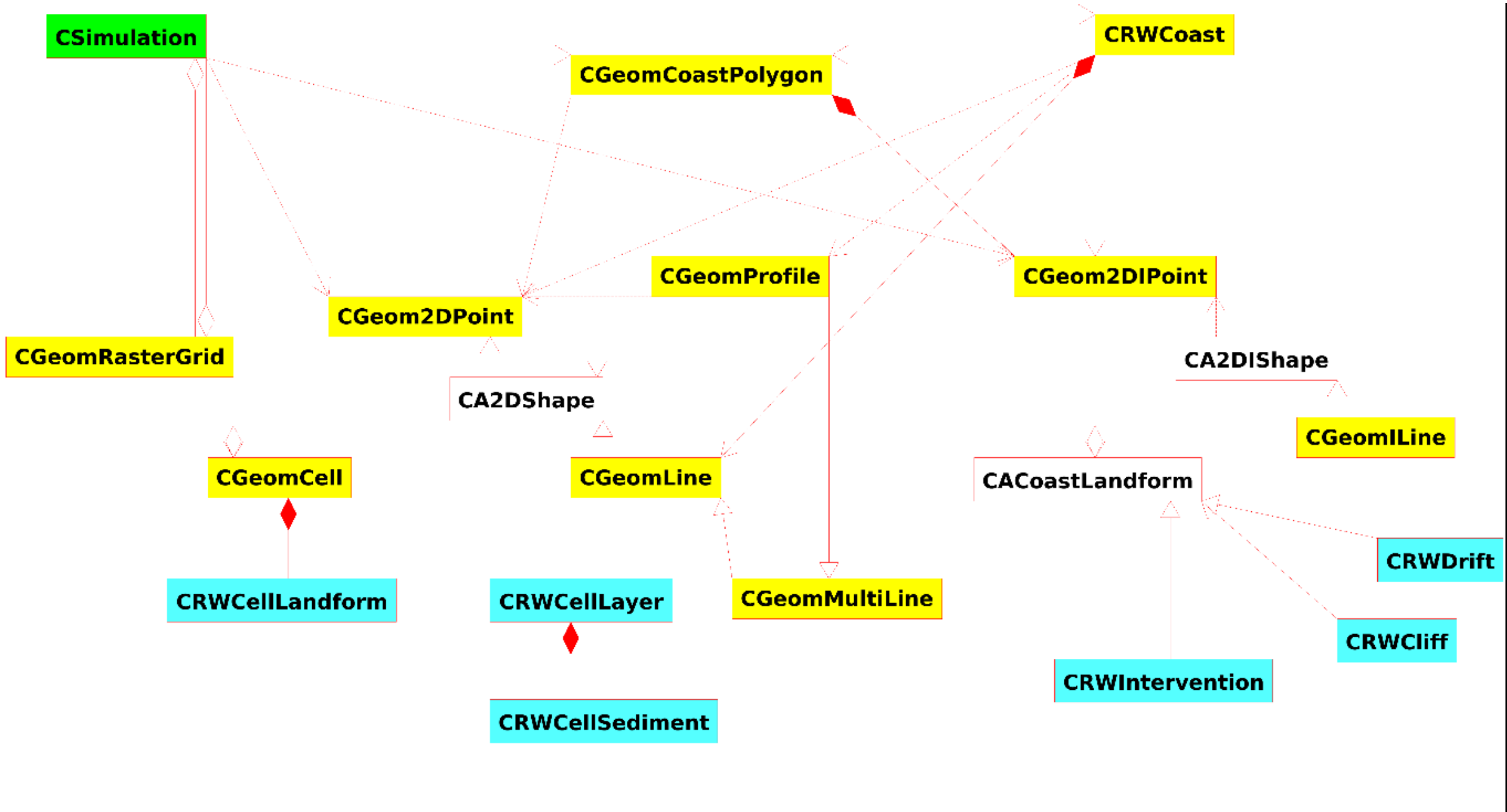

**Figure 13. Class diagram showing the three main classes included in CoastalME. The main class is the Simulation class (green) where the time step operations are defined. A number of abstract classes (yellow) are used to build the different geometries (i.e. vector coastline/profiles, raster coastline/profiles,…). Classes that relates with real geomorphological objects (i.e. cliff, intervention, …) are shown in blue. A full list of clasess and methods included in CoastalME can be found in the framwework documentation (see code availability).**

**Table 1. Pseudo code of CoastalME workflow**

```
Program cme
Initialize Simulation
For (iTimeStep =1; iTimeStep <= MaxNumTimeSteps, iTtimeStep++)
{
        Update External Forcing (SWL and wave properties at deep water)
        Update Human Structural Interventions
        Traces coastline
        Classify landform types
        For any open coast landform
        {
                Trace coastline normal, extract elevation profiles and create sediment-sharing-polygons boundaries
                Calculate wave induce hydrodynamic
                {
                        For each profile first and then interpolate to all grid cells
                        Modify wave properties within shadow zone
                        Get wave properties at each coastline point
                }
                Erode consolidated shore platform and cliff
                        Calculate Potential erosion and then iteratively calculate actual erosion
                        Transfer eroded material to corresponding unconsolidated sediment fraction
                Erode and deposit unconsolidated layer
                        Calculate Potential sediment transport and then iteratively calculate actual sediment transport
                        Update layer thickness on all active grid cells
                Updated DEM becomes the initial DEM for the next time step
}
```

**Table 2. CoastalME adopted ontology of coastal landforms and human interventions [*French et al.*, 2016a]. Underlined the ones already included in version 1.0 of CoastalME.**

| Coastal landforms, hinterland and sediment stores | | | |
|---|---|---|---|
| **Landform** | | **Hinterland** | **Sediment store** |
| Cliff | Inlet channel | High ground | Seabed grave |
| Shore platform | Ebb delta | Low ground | Seabed sand |
| Beach | Flood delta | Reclaimed | Seabed mud |
| Beach ridge | Bank | | Suspended mud |
| Tombolo | Channel | | |
| Dune | Tidal flat | | |
| Spit | Saltmarsh | | |
| Rock outcrop | Brakish marsh | | |
| Lagoon | River | | |
| **Human interventions** | | | |
| **Structural** | **Indicative purpose** | **Non-structural** | **Indicative purpose** |
| Seawall | Erosion protection | Dredging | Navigation; mining |
| Revetment | Erosion protection | Dredge disposal | Spoil disposal |
| Bulkhead | Erosion protection | Sediment recharge | Restoration of sediment deficit |
| Embankment | Flood protection | Sediment bypassing | Continuity of sediment pathway; navigation |
| Barrage | Flood protection | Sediment recycling | Resilience (beach profiling) |
| Breakwater | Wave energy reduction | | |
| Detached breakwater(s) | Wave energy reduction | | |
| Groyne(s) | Sediment retention | | |
| Training wall | Channel stabilisation/navigation | | |
| Jetty | Varied | | |
| Outfall | Drainage/dispersal | | |
| Quay | Navigation/trade | | |
| Dock | Navigation/trade | | |
| Weir | Regulation of river gradient and or tidal limit | | |

**Table 3. Component model, role of component and how is implemented in CoastalME.**

| Component Model | Role | How is implemented |
|---|---|---|
| COVE wave description | Wave propagation | C++ code |
| CSHORE* | | Fortran library |
| CERC | Per-polygon movement of unconsolidated sediment | C++ code |
| Kamphuis | | C++ code |
| Dean profile | Along-profile distribution of unconsolidated sediment | C++ code |

**Table 4. Capabilities of model components SCAPE and COVE and integrated CoastalME composition.**

| Capability | COVE | SCAPE | CoastalME |
|---|---|---|---|
| Soft cliff erosion and beach interaction | N | Y | Y |
| Highly irregular coastlines | Y | N | Y |
| Sediment supply limited environments | N | Y | Y |
| Transport supply limited environments | Y | Y | Y |
| Handle three different sediment fractions | N | N | Y |
| High Angle Wave shoreline instabilities | Y | N | Y[1] |
| Diffraction | Y | N | Y |
| Effects of regional bathymetry on local alongshore sediment transport gradients | N[2] | N[2] | Y |

Y & N yes and no capability included

[1]Implemented but yet to be tested (i.e. need to include wave shadowing and diffraction)

[2]Regional bathymetry can be achieved in an offline manner, driving the model with more sophisticated wave transformation model