# Peer review of "Coastal Modelling Environment version 1.0: a framework for integrating landform-specific component models in order to simulate decadal to centennial morphological changes on complex coasts"

_Geoscientific Model Development, 2016_

## Referee Comment (RC1) · Anonymous Referee #1 · 26 Feb 2017

Referee Comments

General comments

This paper describes a large scale coastal behavioural model intended to allow simulation and forecast of evolution in coastal features over decades for the purposes of coastal management. The model will eventually include various modules representing natural processes and human intervention on a range of coastal features, including cliffs, beaches, inlets, and estuaries, but only beaches and cliffs are represented in the present version.

The paper does not attempt to validate the model against observations; the authors

have explicitly left that task to another paper. Instead, the paper describes the model philosophy and framework, and provides some mechanistic details and results of two test cases.

Sediment-transport models represent processes with a mix of fundamental physics, empiricism, and heuristics. Physics in CoastalME is limited to conservation of mass and wave energy. Empirical formulae are used for longshore transport, and heuristic models are used to represent the beach shape (Dean profiles) and vertical distribution of erosion rates (f1) on cliff faces. The equations for longshore transport rate, beach profiles, and cliff erosion depend on calibration coefficients including Kls, A, R, and the profile f1. These coefficients do not have universal values, but depend on grain-size distributions, assumptions about underlying stratigraphy, rock properties, wave climate, and other site-specific variables that might evolve over time. In addition, CoastalME depends on a number of user-specified parameters that are likely to change results, including the raster cell size, the spacing of shore-normal profiles, the selection of closure depth, timing of cliff collapse, the distribution and relative erodibility of non-cohesive sediment, the depth to non-erodible basement, and others.

The paper uses mostly prose, rather than equations or diagrams, to portray the model mechanics. This makes some of the sections long, sometimes confusing, and ultimately not sufficiently informative. Said another way, it would not be possible to reproduce the model structure or even the fundamental grid / profile / polygon geometry based on this description. Some well-designed diagrams with formulae showing how profiles relate to the raster grid would be helpful. Several of the figures describe aspects of other models, and they could be removed and replaced with diagrams specific to CoastalME.

The authors argue that CoastalME provides an alternative object-oriented approach that combines advantages of both raster-based and vector based structures. The model is a work in progress, and the potential for coupling more landscape objects remains to be achieved. The advantages of the raster-vector combination are not readily

apparent in the two cases presented, and the approach seems to require a lot of iteration and smoothing. Overall, the present model formulation does not appear to be usable for the purpose of informing coastal management, and the paper does only a fair job of describing the model.

The authors deserve great credit for providing open-source code. The model is easy to find on github, and builds easily on Linux. The same is true for SCAPE. The code is well commented and documented with Doxygen. However, I could not fine input files to run a demo cases.

Specific comments

p7 l8 I do not agree that the "most general" way to account for sediment is in a 2DH grid. Maybe you mean "most common". p8 I do not agree with the argument that small cells are required by fast-moving information. Small cells improve resolution, especially of sharp fronts, as long as numerical diffusion is limited. Note that CFL constraints apply to explicit formulations; time steps can be greater than CFL with implicit formulations.

Section 3.1 l17 says the model preferentially locates profiles on capes, but this does not appear to be the case in Fig 9 or Fig 10. As mentioned below, a figure showing how raster cells are associated with profiles and sediment fluxes would be helpful. Does the random spacing of profile change the results? What artefacts are being avoided by doing this?

Section 3.2 If I understand this correctly, wave properties are calculated for each cell, based on properties of the seaward cell, which accounts for local refraction and breaking. I don't think this method conserves global wave energy and allows it to be focused on regions of converging wave-propagation rays (ie., headlands) or away from bays. I think this is evident in Fig 10, where it appears that wave energy is not concentrated on the tips of the cusps. COVE uses an approach to decrease wave energy in shadow zones (but not, as far as I can tell, to concentrate energy on headlands) but this is not yet implemented in CoastalME. In keeping with the modular approach to CoastalME, it

seems like a raster-based wave model like SWAN could be used here.

Section 3.3

Eqns 4 and 5 are bulk transport equations calibrated to the median grain size. It is not clear that they should be applied separately to fractions of the unconsolidated sediment, or what the coefficients would be in that application.

The description of sediment flux and net erosion or deposition is confusing and could be improved with a figure and/or equations (e.g., the discrete version of Eqn. 3, with f defined). It is not clear where the fluxes are located (at profiles or between them) and whether the supply-limited contribution from an eroding cell is ameliorated by contributions to that cell from upstream. It is also not clear how the varying sizes of the polygons are accounted for in f(dQ/ds), because the relationship between dn/dt and elevation changes (or displacement of Dean profiles) depends on the varying polygon areas (or profile lengths). The text at the bottom of p.18 tries to explain how this is done, but does not mention polygon area, and indicates that erosion or deposition is accommodated by changing the profiles at polygon edges, rather than over the entire polygon. It is difficult to see how this can be done in a consistent way that conserves mass, adjusts profiles as grain size changes, and does not produce unrealistic discontinuities. The authors state that two profiles could merge if they if they intersect offshore, and on p19, l13, they indicate that sediment flux is pro-rated according to the shared length of the boundaries. This has the potential transporting sediment among polygons that mate only at depth, bypassing a shallower polygon, which seems unrealistic.

All of this sounds very iterative and ad hoc, but maybe some diagrammed examples would clarify the process.

p. 19 l24-27. Smoothing the grid is a diffusive procedure, as is smoothing the coastline.

Section 3.4

Most modelers use the term periodic, rather than "mobius", to refer to boundary conditions that feed output from one boundary back into the model at the opposite boundary.

Section 3.5

I have not seen an equation for f1 in either this paper or the SCAPE papers I have read. It seems like an interesting heuristic approach, and it would be helpful if the curve of f1 derived from Fig 5b in Walkden and Hall (2005) was specified.

This section describes the cliff erosion process with the prose approach that could be improved with a figure showing how the Dean profile is applied.

References

A few of the references refer to ephemeral sites like Wikipedia that don't always serve as a reliable citation. No DOIs are provided.

Following references are incomplete: Hutton 2014 Payo 2014 Stive 1997 Terwindt, 1990 van Rijn, 2002 Walkden, 2015

Fig. 1 This could be omitted or replaced with a figure that represent the geometry of CoastalME. Fig. 2. This could be omitted or replaced with a similar figure that represents the processes in CoastalME. Fig 3. This could be omitted. Fig 4. This is a key figure that could be improved. One flaw is that it shows profile changes at the depth of closure...I would assume that no changes should occur at or seaward of this depth. A zoomed in figure that shows the relationship between the raster cells, the boxy coastline found byt tracing the raster, the smoothed version of the coastline, the projection of the shore-normal profiles, and the raster cells associated with each profile that "share" sediment with adjacent profiles. Fig 6. This could be omitted. The directional convention is a level of detail needed to make input files, but not to describe the model. Fig 7. This figure could be eliminated. Text in Section 2.3 covers this in better detail. Fig 8. This figure is illegible. It might be useful in a developers guide, but this paper does not deal with the object structure in detail. It could be eliminated. Fig 9. Why are the profiles not parallel in panel b? Why is the spacing so variable

[Figure]

in all panels? Why are there no profiles on the capes in panel d? Can this figure be used to show the association with the raster grid and the polygonal sections? Fig 10. Same questions about profile location and spacing. In addition, the distribution of wave energy does not look right. Wave heights should be highest on the headlands, especially in the 270 deg. case. Fig. 11. This is a good figure. A similar figure showing how the sediment is redistributed to make a Dean profile would be helpful.

Technical corrections Eqn. 1. Missing g in denominator of second term on right Eqn. 4. Kls is not defined. Eqn. 5. the coefficient 2.33 assumes seawater density of 1030 kg/m3 (van Rijn, 2005), not the value of 1025 kg/m3 specified on l16. Eqn. 6. the dimensions in this equation don't work out...the right side has dimensions of m-2 s-1...so volume transport per meter width. It might be good to define immersed weight transport and show the relationships between I, volume transport, and mass trasport. Eqn. 8. The slope should be dzs/dys or tan(alpha), but not tan(dzs/dys). Eqn. 9. Same comment.

Typos p2 l20 pool of well-understood open-access models... p4 l4 models (Murray... p4 l27 Volumetric model(s) represent . . . p5 l5 COVE is inspired by the Coastal Evolution Model. . . p6 l11...geometrically constrained by human interventions p7 l3 Sediment is stored as . . . or in suspension p7 l9 last phrase does not seem to make sense p8 l17 Lewy Condition p25, l4 all three

---

## Referee Comment (RC2) · Anonymous Referee #2 · 2 Mar 2017

GENERAL COMMENTS

This paper proposes a new method to simulate the evolution of decadal to centennial morphological changes and could be use to help decision making in coastal management studies. The approach is based on a new modular framework that links independent software together such as the one implemented in OpenMI.

Here the focus is on large-scale coastal evolution and a great development effort has consisted in transposing the behavioural rules of each independent model together in order to pass the information from one model to another. Three models are currently coupled within the CoastalME framework namely: COVE, SCAPE and ASMITA. A great emphasis is made in the manuscript on the first 2 models whereas the integration of

ASMITA model with the two others is lacking details and illustrations.

The framework development mainly consists in passing information between each model spatial reference system and consequently involves a lot of geometrical calculations that are fairly well described in the paper. For any given iteration, the model outputs are then stored on raster grid used by the authors as the main spatial representation for their framework. In my view, some parts of the paper in regards to how the model conserves volume through the successive interpolation and smoothing functions is still unclear and will need more explanations.

The authors do not provide any validation examples of their new framework in this paper as it is left for another paper. Two examples are however described in the last section of the manuscript but I found them not really illustrative of the framework capability, as they do not address the problem of decadal to centennial morphological changes that is what this work is about. I also think that these examples will need to be reworked quite extensively to be more appropriate for publication.

SPECIFIC COMMENTS

+ Page 2 line 18: space missing "in particular be theOpen Modelling Interface…"

+ Page 4 line 4: space missing "specific models(Murray.."

+ Page 4 line 16: change "provide as a significant " to "provide a significant "

+ Page 4 line 24: change "meso-escale" to "meso-scale"

+ Page 6 line 16: missing ) after Wadden Sea).

+ Page 8 line 4 to 6: indeed CA models have been used on regular grid but I don't see why you are citing them here most current hydrodynamic/sediment transport models are based on gridded spatial discretisation. I will just delete this sentence…

+ Page 8 line 14: delete "More seriously,"

+ Page 8 line 17: space missing "-LewyCondition"

+ Page 9 line 10: Model output of the model  consists. . .

+ Page 10 line 27: "SWL can be fixed or assumed to change linearly every time step", does the user input a sea-level curve and SWL is linearly interpolated based on this curve ? If this is the case you will need to make it clearer in this paragraph.

+ Page 11 lines 3 & 14 and page 12 line 14: how do you ensure mass conservation when using smoothing algorithm?

+ Page 11 line 15: how to you set the profiles on the grid edge and why aren't they normal to the coastline? Additional explanation is required. . .

+ Page 13 line 15: instead of "(code availability )" provide the link to the configuration file (maybe using a shorten url: (http://www.coastalme.org.uk/doku/doku.php?id=inputs_and_outputs:myinputs.dat)

+ Page 13 line 29: again instead of code availability, provide a direct link to where this information can be found on the web.

+ Page 14/15 1ines 25 to 29 and 1 to 3: you should reference Figure 4 in this section to make it more clear to the reader and add some of the defined notations to Figure 4.

+ Page 15 line 12/13: smoothing the coastal profiles require an additional step in the computation in comparison to the method implemented in COVE, you should explain how the resulting curvature calculation improves the prediction of alongshore sediment transport algorithm in the context of CoastalME.

+ Page 16 line 13: you should provide the equation for the downwearing erosion $\varepsilon$

+ Page 16 line 25: Is it possible to set some spatially variable active layer availability factors for each sediment? How is the availability factor for the active layer related to the active layer through time? For example let assumes that $\alpha$  is set to 0 for a given sediment type and that through simulation time steps some of these sediments start to

be deposited, does it mean that they will never been eroded away. I guess this is not the case but it requires better explanation in this section.

+ Pages 17 to 19 section 3.3 will need to be shortened and will greatly be improved with a figure or diagram to help readers. This is really important as it defines the alongshore transport algorithm. Something similar to what is done in section 3.5 with Figure 11 will be really helpful.

+ Page 21 line 28: "is considered to have its base a user-specified depth d1 below the SWL, " this sentence doesn't read well and needs to be rewritten.

+ Page 23 line 6: is the sediment porosity depth dependent, in other word do you account for compaction of sediment with time? I guess this could be important considering that the code is designed to look at centennial morphological changes.

+ Page 23 line 8: like comments above I will suggest that instead of code availability, you should provide a direct link to where this information can be found on the web.

+ Page 23 lines 10 to 27: you only mentioned the duration of your model in the Figure 12 caption this needs to be provided in the text as well. The purpose of this new code is to work at decadal to centennial scale I understand that you are planning to do a second paper but it will be good to have an example that is relevant to this scale in this paper. I would also like to see the full DEM result as well and not only a small part of it in Figure 12. You should also provide the time involved to simulate this 1-year morphological evolution so people can judge on the performance of the code.

+ Page 24 line 1: COVE2015 needs to be changed to COVE.

+ Page 24 lines 11 to 19: there is no mention of the settings of this experiment and of its duration. . . you wrote a "sufficiently long time" you will need to provide some numbers here. I think this part needs to be more developed. At the moment the description and interpretations of the results are lacking.

+ Page 25 lines 7 to 9: you will need to provide some metrics of the code efficiency and

CPU time to make it clearer in your manuscript. Page 25 lines 10 to 12: can you be more specific are you referring to a possible alternative to model like CAESAR? This last sentence will need to give the reader a better idea of how this could be done more references or should be deleted.

REFERENCES

Several of the references are from Wikipedia, you should provide citable references instead. In several parts of the manuscript (as pointed above) you refer to "code availability"instead you should provide the link to the places where the reader can find the information on the code website. I found 3 references which are not complete and are missing journal names or editors: - Hutton 2014 - Stive 1997 - van Rijn, 2002

FIGURES

+ Page 35 Figure 5 will need a color bar for the DEM on the left side and the resolution of this figure will need to be improved

+ Page 36 Figure 6 line 4-5: change '"this" (white dot) coastline node" to "the considered coastline node (white dot)".

+ Page 37 & 38 Figures 7 & 8: resolution needs to be improved

+ Page 39 & 40 Figures 9 & 10: resolution of the text parts needs to be improved. In Figure 10 I think you should not plot the breaking wave height legend it is confusing as there is no colored dot on the figure itself.
* * *

---

## Author Comment (AC1) · 13 Apr 2017

The authors appreciate the comments and feedbacks provided by Reviewer #1. Please find below a detailed description about how the authors has addressed both the general and detailed reviewer's comment on the reviewed manuscript. (R = reviewer comment; A = authors response)

General comments

R: Sediment-transport models represent processes with a mix of fundamental physics, empiricism, and heuristics. Physics in CoastalME is limited to conservation of mass and wave energy. Empirical formulae are used for longshore transport, and heuristic

models are used to represent the beach shape (Dean profiles) and vertical distribution of erosion rates (f1) on cliff faces. The equations for longshore transport rate, beach profiles, and cliff erosion depend on calibration coefficients including Kls, A, R, and the profile f1. These coefficients do not have universal values, but depend on grain-size distributions, assumptions about underlying stratigraphy, rock properties, wave climate, and other site-specific variables that might evolve over time. In addition, CoastalME depends on a number of user-specified parameters that are likely to change results, including the raster cell size, the spacing of shore-normal profiles, the selection of closure depth, timing of cliff collapse, the distribution and relative erodibility of noncohesive sediment, the depth to non-erodible basement, and others.

A: It is important to emphasize that CoastalME is not a model it is a framework for building coupled Large-Scale Coastal Behavioural (LSCB) models. This manuscript describes the elements of the framework and, as a proof of concept, how the framework may be used to integrate two models: one for cliff-beach interaction (SCAPE) and one for alongshore sediment transport (COVE). With this in mind, the reviewer's general comments are both perceptive and encouraging since they recognise that the CoastalME framework satisfies some of the fundamental requirements of any LSCB model (i.e. mass conservation and sediment transport driven by wave energy), and also enables representation of the temporal and spatial variability of the attributes that influence coastal change (i.e. sediment size, stratigraphy, rock properties, wave climate...). We fully acknowledge that (as with all non-reductionist models) some user-specified input parameters will influence results. A sensitivity analysis of the framework is planned. This will assist us in selecting sensible default values of these parameters for users, and in advising users on the limits of applicability of these values.

R: The paper uses mostly prose, rather than equations or diagrams, to portray the model mechanics. This makes some of the sections long, sometimes confusing, and ultimately not sufficiently informative. Said another way, it would not be possible to reproduce the model structure or even the fundamental grid / profile / polygon geometry

based on this description. Some well-designed diagrams with formulae showing how profiles relate to the raster grid would be helpful. Several of the figures describe aspects of other models, and they could be removed and replaced with diagrams specific to CoastalME.

A: We acknowledge that the figures should be both improved and made more specific to CoastalME; this has been done on the revised version of the manuscript. Regarding the description of 'model mechanics': again, it is important to emphasise that CoastalME is a framework, not a model. The novel element of our work is in developing a framework which is capable of integrating component models. The component models need not themselves be novel, and indeed we have chosen here to integrate two well-documented component models: SCAPE and COVE. We have supplied equations which capture the essential characteristics of each component model. But it is not at all clear how one might use equations to describe the integrating framework. We have, instead improved the description of the framework mainly by improving the figures, in particular showing how coast, profiles, and polygons relate to the raster grid and how they are dynamically updated every time step. Additionally, the source code of the CoastalME framework is both freely available and well documented (see below). The combination of the revised manuscript and the source code documentation will provide a comprehensive description of the framework's structure.

R: The authors argue that CoastalME provides an alternative object-oriented approach that combines advantages of both raster-based and vector based structures. The model is a work in progress, and the potential for coupling more landscape objects remains to be achieved. The advantages of the raster-vector combination are not readily apparent in the two cases presented, and the approach seems to require a lot of iteration and smoothing. Overall, the present model formulation does not appear to be usable for the purpose of informing coastal management, and the paper does only a fair job of describing the model.

A: CoastalME is a novel framework for integrating component models: it is not a

model. Raper and Livingstone (1995) suggested that the next step in model integration should be a spatial representation of component models within an object-oriented environment, rather than an ad-hoc integration of incompatible systems, which inevitably forces representational compromises. Here, we have demonstrated how the concepts behind the cliff-beach interaction in SCAPE and the shoreline response to changes in alongshore sediment transport gradients in COVE can be integrated in CoastalME. We do not claim that the integrated SCAPE-COVE is currently usable for the purpose of informing coastal management. In this paper, we limit ourselves to a description of the framework's structure and the philosophy which underpins it, together with some results from a linkage of SCAPE and COVE as proof of concept. We will aim in subsequent publications to demonstrate the potential of CoastalME's linked raster-vector approach for informing coastal management. Please note, too, that smoothing is used only when tracing the coastline and when detemining slope gradients on profiles. Mass is fully conserved within the CoastalME framework. For more regarding the description of the framework, see below. Raper, J. and D. Livingstone (1995). "Development of a geomorphological spatial model using object-oriented design." International Journal of Geographical Information Systems 9(4): 359-383.

R: The authors deserve great credit for providing open-source code. The model is easy to find on github, and builds easily on Linux. The same is true for SCAPE. The code is well commented and documented with Doxygen. However, I could not fine input files to run a demo cases. A: The authors are grateful for the reviewer's acknowledgement of the value of open-sourcing CoastalME's code, and providing Doxygen documentation. The inputs and outputs used for the test cases can be found at http://www.channelcoast.org/iCOASST/COASTAL_ME/. This is now clearly stated in the revised manuscript. Finally, regarding the earlier comments on reproducibility and adequacy of the description of the CoastalME framework: we suggest that the free availability of the CoastalME code, and the effort that has gone into documenting this code, are both relevant. The CoastalME source code is itself intended to be part of the framework's description. The free availability of the source code means that any

scientist who wishes to learn more about the CoastalME framework, or to modify it, is easily able to do so.

Detailed comments (L#: Line number)

R: p7 l8 I do not agree that the "most general" way to account for sediment is in a 2DH grid. Maybe you mean "most common". p8 I do not agree with the argument that small cells are required by fast-moving information. Small cells improve resolution, especially of sharp fronts, as long as numerical diffusion is limited. Note that CFL constraints apply to explicit formulations; time steps can be greater than CFL with implicit formulations.

A: "Most general" has been replaced with "most common" on the revised manuscript. The note on CFL constraints has been also added for completeness.

R: Section 3.1 l17 says the model preferentially locates profiles on capes, but this does not appear to be the case in Fig 9 or Fig 10. As mentioned below, a figure showing how raster cells are associated with profiles and sediment fluxes would be helpful. Does the random spacing of profile change the results? What artefacts are being avoided by doing this?

A: A bug on the cape-normal allocation has been fixed and Figures 9 and 10 have been re-edited. A new figure showing how raster cells are associated with profiles and sediment fluxes has been added (new Figure x). Coastline profiles can be allocated at a deterministic alongshore spacing or allow some randomness of the spacing. For the test cases presented here, using deterministic or random spacing has no effect on the results. Having the random option allows the user quickly assess the sensitivity of the results to user-defined parameters such as profile spacing. Profiles are conceptual constructs that we use to simulate complex 3D open systems. The polygon averaged properties (i.e. grain size, rock resistance,...) will vary with the profile distance and will influence the among of sediment being eroded/transported between polygons. We acknowledge that this might create artefacts but this will be likely site-specific.

R: Section 3.2 If I understand this correctly, wave properties are calculated for each cell, based on properties of the seaward cell, which accounts for local refraction and breaking. I don't think this method conserves global wave energy and allows it to be focused on regions of converging wave-propagation rays (ie., headlands) or away from bays. I think this is evident in Fig 10, where it appears that wave energy is not concentrated on the tips of the cusps. COVE uses an approach to decrease wave energy in shadow zones (but not, as far as I can tell, to concentrate energy on headlands) but this is not yet implemented in CoastalME. In keeping with the modular approach to CoastalME, it seems like a raster-based wave model like SWAN could be used here.

A: The cape-normal allocation algorithm purposes is in place to ensure that wave energy at the capes is well captured using the simple linear wave theory. We have fixed the bug on the cape-normal allocation (see previous comment and new Figures 9 and 10) and wave energy focusing on capes is now clear. For completeness with COVE, we have also implemented the simple refraction and shadowing rules included in COVE. The reviewer is correct in noticing that a raster-based wave component model like SWAN could be replace the simple linear wave propagation component model that we have implemented. This modularity is one of the strengths of CoastalME.

R: Section 3.3. Eqns 4 and 5 are bulk transport equations calibrated to the median grain size. It is not clear that they should be applied separately to fractions of the unconsolidated sediment, or what the coefficients would be in that application. The description of sediment flux and net erosion or deposition is confusing and could be improved with a figure and/or equations (e.g., the discrete version of Eqn. 3, with f defined). It is not clear where the fluxes are located (at profiles or between them) and whether the supply-limited contribution from an eroding cell is ameliorated by contributions to that cell from upstream. It is also not clear how the varying sizes of the polygons are accounted for in $f(dQ/ds)$, because the relationship between $dn/dt$ and elevation changes (or displacement of Dean profiles) depends on the varying polygon areas (or profile lengths). The text at the bottom of p.18 tries to explain how this is

done, but does not mention polygon area, and indicates that erosion or deposition is accommodated by changing the profiles at polygon edges, rather than over the entire polygon. It is difficult to see how this can be done in a consistent way that conserves mass, adjusts profiles as grain size changes, and does not produce unrealistic discontinuities. The authors state that two profiles could merge if they if they intersect offshore, and on p19, l13, they indicate that sediment flux is pro-rated according to the shared length of the boundaries. This has the potential transporting sediment among polygons that mate only at depth, bypassing a shallower polygon, which seems unrealistic. All of this sounds very iterative and ad hoc, but maybe some diagrammed examples would clarify the process.

A: The bulk alongshore sediment transport is calculated using a median grain size and not for each sediment fraction independently. This has now been clearly stated in the revised manuscript. The description of sediment flux and net erosion or deposition has been improved with a new figure that illustrates how the potential erosion for each polygon is calculated (similar to Figure 11 that has been acknowledged by this reviewer to be a good figure) and we have made the equations explicit. Mass is conserved by ensuring that all shared sediments between polygons is either deposited on the polygon cells or keep it in suspension. As stated in p21 l 22 "Checks are performed during this loop to ensure that cells are not eroded several times and that there are no cells within the active zone of this polygon at which the actual erosion potential has not been calculated". In respect to the comment about sediment bypassing only at depth, we advise that on concave coastlines (which are commonly found) polygons meet at relatively shallow water (i.e. well within the active zone) and it is therefore realistic to share sediments between polygons.

R: p. 19 l24-27. Smoothing the grid is a diffusive procedure, as is smoothing the coastline. A: Grid smoothing is not needed in the current version of the CoastalME framework, and the grid smoothing routine has been removed from the updated code. It is important to notice that, the coastline is only smoothed to draw the profiles. The

start of the coastline profiles is the raster cell identified as a coastal point by intersecting the water level at each time step with the DEM. Coastline smoothing ensures that each profile's planform orientation is not unrealistic, as would be the case with an unsmoothed coastline directly constructed from the discrete cells of the raster DEM.

R: Section 3.4. Most modellers use the term periodic, rather than "mobius", to refer to boundary conditions that feed output from one boundary back into the model at the opposite boundary. A: Mobius has been replaced by periodic as suggested, in the revised manuscript.

R: Section 3.5. I have not seen an equation for f1 in either this paper or the SCAPE papers I have read. It seems like an interesting heuristic approach, and it would be helpful if the curve of f1 derived from Fig 5b in Walkden and Hall (2005) was specified. This section describes the cliff erosion process with the prose approach that could be improved with a figure showing how the Dean profile is applied.

A: A new figure showing how the Dean profile is applied is now included (Figure X). A look up table (i.e. instead of a function) has been used to replicate the Fig5b in Walkden and Hall (2005). The values used to represent the shape function has been included as a new Table X. R: References. A few of the references refer to ephemeral sites like Wikipedia that don't always serve as a reliable citation. No DOIs are provided. Following references are incomplete: Hutton 2014 Payo 2014 Stive 1997 Terwindt, 1990 van Rijn, 2002 Walkden, 2015.

A: References to Wikipedia has been replaced by peer reviewed papers and DOIs included where missing. The above-cited reference has been amended on the revised manuscript.

R: Fig. 1 This could be omitted or replaced with a figure that represent the geometry of CoastalME. Fig. 2. This could be omitted or replaced with a similar figure that represents the processes in CoastalME. Fig 3. This could be omitted. Fig 4. This is a key figure that could be improved. One flaw is that it shows profile changes at

the depth of closure...I would assume that no changes should occur at or seaward of this depth. A zoomed in figure that shows the relationship between the raster cells, the boxy coastline found by tracing the raster, the smoothed version of the coastline, the projection of the shore-normal profiles, and the raster cells associated with each profile that "share" sediment with adjacent profiles. Fig 6. This could be omitted. The directional convention is a level of detail needed to make input files, but not to describe the model. Fig 7. This figure could be eliminated. Text in Section 2.3 covers this in better detail. Fig 8. This figure is illegible. It might be useful in a developers guide, but this paper does not deal with the object structure in detail. It could be eliminated. Fig 9. Why are the profiles not parallel in panel b? Why is the spacing so variable in all panels? Why are there no profiles on the capes in panel d? Can this figure be used to show the association with the raster grid and the polygonal sections? Fig 10. Same questions about profile location and spacing. In addition, the distribution of wave energy does not look right. Wave heights should be highest on the headlands, especially in the 270 deg. case. Fig. 11. This is a good figure. A similar figure showing how the sediment is redistributed to make a Dean profile would be helpful.

A: Fig. 1 has been replaced with a figure that better represents the geometry of CoastalME. Fig 2 has been re-edited to more clearly illustrate how is represented in CoastalME. Fig. 3 has been omitted. Fig. 4 has been improved to show the relationship between the raster cells, the boxy coastline found by directly tracing the raster, the smoothed version of the coastline, the projection of the shore-normal profiles, and the raster cells associated with each profile that "share" sediment with adjacent profiles. Fig. 6 has not been omitted for completeness. This manuscript is intended to be the key descriptive reference for the CoastalME framework and so, in the authors' experience, it is good practice to explicitly show the direction convention used. Fig. 7 and Fig. 8 has been omitted. Fig 9 and Fig 10 has been re-edited after fixing the bug on the cape allocation routine. A new figure X, showing how the sediment is redistributed to make a Dean profile, has been added.

R: Technical corrections Eqn. 1. Missing g in denominator of second term on right Eqn. 4. Kls is not defined. Eqn. 5. the coefficient 2.33 assumes seawater density of 1030 kg/m3 (van Rijn, 2005), not the value of 1025 kg/m3 specified on l16. Eqn. 6. The dimensions in this equation don't work out...the right side has dimensions of m-2 s-1...so volume transport per meter width. It might be good to define immersed weight transport and show the relationships between I, volume transport, and mass trasport. Eqn. 8. The slope should be dzs/dys or tan(alpha), but not tan(dzs/dys). Eqn. 9. Same comment.

A: Missing g in Eq (1) has been added. The definition of Kls in Eq 4 is now included. The density value on l16 has been changed to 1030 kg/m3. Eqn. 6 is volume transport per meter width. The relationship between the immersed weight transport and mass transport are found elsewhere (i.e. van Rijn, 2002) and therefore not included. The slope has been expressed as dzs/dys in Eqn 8 and 9.

R: Typos p2 l20 pool of well-understood open-access models... p4 l4 models (Murray... p4 l27 Volumetric model(s) represent . . . p5 l5 COVE is inspired by the Coastal Evolution Model. . . p6 l11...geometrically constrained by human interventions p7 l3 Sediment is stored as . . . or in suspension p7 l9 last phrase does not seem to make sense p8 l17 Lewy Condition p25, l4 all three

A: The above typos have been amended on the reviewed document.

Please also note the supplement to this comment:
http://www.geosci-model-dev-discuss.net/gmd-2016-264/gmd-2016-264-AC1-supplement.pdf

———————————————

---

## Author Comment (AC2) · 13 Apr 2017

The authors appreciate the comments and feedbacks provided by Reviewer #2. Please find below a detailed description about how the authors has addressed both the general and detailed reviewer's comment on the reviewed manuscript. (R = reviewer comment; A = authors response)

General comments

R: This paper proposes a new method to simulate the evolution of decadal to centennial morphological changes and could be use to help decision making in coastal management studies. The approach is based on a new modular framework that links

independent software together such as the one implemented in OpenMI.

Here the focus is on large-scale coastal evolution and a great development effort has consisted in transposing the behavioural rules of each independent model together in order to pass the information from one model to another. Three models are currently coupled within the CoastalME framework namely: COVE, SCAPE and ASMITA. A great emphasis is made in the manuscript on the first 2 models whereas the integration of ASMITA model with the two others is lacking details and illustrations.

The framework development mainly consists in passing information between each model spatial reference system and consequently involves a lot of geometrical calculations that are fairly well described in the paper. For any given iteration, the model outputs are then stored on raster grid used by the authors as the main spatial representation for their framework. In my view, some parts of the paper in regards to how the model conserves volume through the successive interpolation and smoothing functions is still unclear and will need more explanations. The authors do not provide any validation examples of their new framework in this paper as it is left for another paper. Two examples are however described in the last section of the manuscript but I found them not really illustrative of the framework capability, as they do not address the problem of decadal to centennial morphological changes that is what this work is about. I also think that these examples will need to be reworked quite extensively to be more appropriate for publication.

A: We are grateful that this reviewer acknowledges that CoastalME is a framework for model linkage, and not a model per se. However, CoastalME does not "link. . . independent software [. . . like. . .OpenMI]". Instead, it links an implementation of the concepts which comprise the component models, and not the component models themselves. We do this in order to avoid those incompatibilities of spatial and temporal representation, and of assumptions regarding coastal process, which bedevil OpenMI-type approaches. As stated more than two decades ago by Raper and Livingstone (1995), the next step in model integration should be a fusion of models in a common temporal and spatial representation within anobject-oriented environment which corresponds (at least in part) to the objects comprising the real-world system which is being represented. The OpenMI approach which aims to integrate discrete incompatible systems, each of which makes assumptions regarding time, space and process, inevitably forces representational compromises. Here, we have demonstrated how the concepts which underpin two component models (cliff-beach interaction in SCAPE and the shoreline response to changes in alongshore sediment transport gradients in COVE) maybe implemented and integrated in CoastalME. ASMITA is a component model for estuaries that is not yet implemented in CoastalME.

In the revised paper, we have added new figures and have streamlined the text to better illustrate the CoastalME framework. Please note that smoothing is used only when tracing the coastline and when determining slope gradients on profiles. Mass is fully conserved within the CoastalME framework.

In this paper, we limit ourselves to a description of the framework's structure and the philosophy which underpins it, together with some results from a linkage of SCAPE and COVE as proof of concept. The test examples are an illustration of how the resulting integrated framework has extended the capabilities of the component models. We will aim in subsequent publications to demonstrate the potential of CoastalME's linked raster-vector approach for informing coastal management. Raper, J. and D. Livingstone (1995). "Development of a geomorphological spatial model using object-oriented design." International Journal of Geographical Information Systems 9(4): 359-383.

Detailed comments (L#: Line number)

R: Page 2 line 18: space missing "in particular be the Open Modelling Interface. . ."

A: Amended.

R: Page 4 line 4: space missing "specific models (Murray"

A: Amended

R: Page 4 line 16: change "provide as a significant " to "provide a significant "

A: Amended.

R: Page 4 line 24: change "meso-escale" to "meso-scale"

A: Amended.

R: Page 6 line 16: missing ) after Wadden Sea).

A: Added.

R: Page 8 line 4 to 6: indeed CA models have been used on regular grid but I don't see why you are citing them here most current hydrodynamic/sediment transport models are based on gridded spatial discretisation. I will just delete this sentence. . .

A: The above sentence has not been deleted. The authors argue that, as the text stands, it provides an easy-to-follow narrative.

R: Page 8 line 14: delete "More seriously,"

A: Deleted.

R: Page 8 line 17: space missing "-LewyCondition"

A: Added.

R: Page 9 line 10: Model output of the model consists. . .

A: Amended

R: Page 10 line 27: "SWL can be fixed or assumed to change linearly every time step", does the user input a sea-level curve and SWL is linearly interpolated based on this curve? If this is the case you will need to make it clearer in this paragraph.

A: At present only the value of the SWL at the end of the simulation is provided as a user input. This is now made it clear on the revised manuscript.

Interactive
comment

R: Page 11 lines 3 & 14 and page 12 line 14: how do you ensure mass conservation when using smoothing algorithm?

A: Grid smoothing is not now needed. The grid smoothing routinehas been removed from the updated code and is no longer mentioned from the manuscript. It is important to notice that the coastline is smoothed only in order to draw the coastline-normal profiles. The start of the coastline profiles is the raster cell identified as a coastal point by intersecting the water level at each time step with the DEM. This coastline smoothing ensures that the planform orientation of the coastline-normal profiles is not unrealistic, as would be the case with an unsmoothed coastline traced directly from the discrete cells of the raster DEM.

R: Page 11 line 15: how to you set the profiles on the grid edge and why aren't they normal to the coastline? Additional explanation is required. . .

A: Grid-edge profiles must run along the edge of the grid. So unless the coastline intersects the grid edge exactly at a right angle, the grid-edge profiles will not be normal to the coastline.

R: Page 13 line 15: instead of "(code availability )" provide the link to the configuration file (maybe using a shortened url: (http://www.coastalme.org.uk/doku/doku.php?id=inputs_and_outputs:myinputs.dat)

A: Instead of adding the urls within the main text, the code availability section has been re-edited and all important urls have been made explicit.

R: Page 13 line 29: again instead of code availability, provide a direct link to where this information can be found on the web.

A: see response to comment above.

R: Page 14/15 1ines 25 to 29 and 1 to 3: you should reference Figure 4 in this section to make it more clear to the reader and add some of the defined notations to Figure 4.

A: Reference to Figure 4 has been included in the revised manuscript and the whole document structure re-organized to make it more clear to the reader.

R: Page 15 line 12/13: smoothing the coastal profiles require an additional step in the computation in comparison to the method implemented in COVE, you should explain how the resulting curvature calculation improves the prediction of alongshore sediment transport algorithm in the context of CoastalME.

A: In COVE the coastline is made of a relatively small number of discrete nodes while in CoastalME the coastline is made of a considerably greater number of discrete raster cells. Thus in CoastalME there are many more coastal points between two polygon boundaries than in COVE. Rather than favouring any single raster point on the coastline, we have used a smoothed coastline to trace the profiles. This smoothed coastline is conceptually equivalent to the use of adjacent nodes in COVE.

R: Page 16 line 13: you should provide the equation for the downwearing erosion "

A: The downwearing erosion is equation is explained in detail in section 3.5. This has been now clearly stated on the revised manuscript.

R: Page 16 line 25: Is it possible to set some spatially variable active layer availability factors for each sediment? How is the availability factor for the active layer related to the active layer through time? For example let assumes that is set to 0 for a given sediment type and that through simulation time steps some of these sediments start to be deposited, does it mean that they will never been eroded away. I guess this is not the case but it requires better explanation in this section.

A: At present the availability factor in the active layer is constant and uniform spatially but there is nothing preventing the user modifying the CoastalME framework to ensure that this value actually varies spatially and over time. Neither SCAPE or COVE has an active layer concept but in CoastalME input file we allow the user to define different erodibilities to the different sediment fractions.

R: Pages 17 to 19 section 3.3 will need to be shortened and will greatly be improved with a figure or diagram to help readers. This is really important as it defines the alongshore transport algorithm. Something similar to what is done in section 3.5 with Figure 11 will be really helpful.

A: The whole document has been re-organized and shortened and more illustrative figures included in the revised manuscript.

R: Page 21 line 28: "is considered to have its base a user-specified depth d1 below the SWL, " this sentence doesn't read well and needs to be rewritten.

A: The sentence has been rewritten as The base of the cliff notch is considered to be at a user-specified depth d1 below the SWL, and the notch is considered to be eroded a length L1 inland (Fig. 11a). ""

R: Page 23 line 6: is the sediment porosity depth dependent, in other word do you account for compaction of sediment with time? I guess this could be important considering that the code is designed to look at centennial morphological changes.

A: Compaction is not considered by either COVE or SCAPE and therefore is assumed constant in CoastalME. There is nothing preventing the user adding a rule to the raster cell class that modifies the porosity over time.

R: Page 23 line 8: like comments above I will suggest that instead of code availability, you should provide a direct link to where this information can be found on the web.

A: Instead of adding the urls within the main text, the code availability section has been re-edited and all important urls have been made explicit.

R: Page 23 lines 10 to 27: you only mentioned the duration of your model in the Figure 12 caption this needs to be provided in the text as well. The purpose of this new code is to work at decadal to centennial scale I understand that you are planning to do a second paper but it will be good to have an example that is relevant to this scale in this paper. I would also like to see the full DEM result as well and not only a small

part of it in Figure 12. You should also provide the time involved to simulate this 1-year morphological evolution so people can judge on the performance of the code.

A: The duration of the model simulations relative to the Figure 12 has been included in the main text of the revised manuscript. Both COVE and SCAPE are models developed to work at decadal to centuries time scale. CoastalME it is a framework that integrate these two models and therefore is by definition applicable to decadal to centuries time scale. We have increased the spatial extent of the test cases from a small 1000m x 500m domain to a larger 100 km x 20 km domain to better illustrate the applicability of the proposed framework to large scale simulations. The performance of these new test cases has now been included in the revised manuscript.

R: Page 24 line 1: COVE2015 needs to be changed to COVE.

A: Amended

R: Page 24 lines 11 to 19: there is no mention of the settings of this experiment and of its duration. . . you wrote a "sufficiently long time" you will need to provide some numbers here. I think this part needs to be more developed. At the moment the description and interpretations of the results are lacking.

A: See above the response to comment on Page 23 lines 10 to 27.

R: Page 25 lines 7 to 9: you will need to provide some metrics of the code efficiency and CPU time to make it clearer in your manuscript.

A: Metrics of code efficiency and CPU time (i.e. ratio of simulation duration and simulated time) has been included on the revised manuscript.

R: Page 25 lines 10 to 12: can you be more specific are you referring to a possible alternative to model like CAESAR? This last sentence will need to give the reader a better idea of how this could be done more references or should be deleted.

A: Yes, the proposed modelling approach is an alternative to cellular landscape evolution models such as CAESAR (Coulthard et al., 2002) and RILL-GROW (Favis-Mortlock et al. 2000). This has now been clearly stated in the revised manuscript and the references below added. Coulthard, T. J., Macklin, M. G., Kirkby, M. J., 2002. A cellular model of Holocene upland river basin and alluvial fan evolution. Earth Surface Processes and Landforms, 27, 3: 269–288. 10.1002/esp.318 Favis-Mortlock, D.T., Boardman, J., Parsons, A.J. and Lascelles, B. (2000). Emergence and erosion: a model for rill initiation and development. Hydrological Processes 14(11-12), 2173-2205.

R: REFERENCES. Several of the references are from Wikipedia, you should provide citable references instead. In several parts of the manuscript (as pointed above) you refer to "code availability" instead you should provide the link to the places where the reader can find the information on the code website. I found 3 references which are not complete and are missing journal names or editors: - Hutton 2014 - Stive 1997 - van Rijn, 2002

A: References to Wikipedia has been replaced by peer reviewed papers and DOIs included where missing. The above cited references has been amended on the revised manuscript.

FIGURES

R: Page 35 Figure 5 will need a color bar for the DEM on the left side and the resolution of this figure will need to be improved

A: colour bar added and resolution increased to 600ppi.

R: Page 36 Figure 6 line 4-5: change '"this" (white dot) coastline node" to "the considered coastline node (white dot)".

A: Amended as suggested.

R: Page 37 & 38 Figures 7 & 8: resolution needs to be improved

A: Resolution increased to 600ppi.

R: Page 39 & 40 Figures 9 & 10: resolution of the text parts needs to be improved. In Figure 10 I think you should not plot the breaking wave height legend it is confusing as there is no colored dot on the figure itself.

A: Resolution of the text parts on Figures 9 & 10 has been increased. There are coloured dots along the shoreline representing the breaking wave height but might be confused with the wave height colours. A different colour scheme has been selected for the breaking wave height on the revised manuscript.

Please also note the supplement to this comment:
http://www.geosci-model-dev-discuss.net/gmd-2016-264/gmd-2016-264-AC2-supplement.pdf

---

## Author Response (AR2)

Dr Thomas Poulet
thomas.poulet@csiro.au
Handling Topical Editor GMD

**Keyworth**

Environmental Science Centre
Keyworth
Nottingham
United Kingdom
NG12 5GG

Telephone  +44(0)115 9363100
Direct Line  +44(0)115 9363103
E-mail  agarcia@bgs.ac.uk
Web  www.bgs.ac.uk

08 June 2017

Dear Dr Poulet,

Please find attached the revised manuscript entitled " COASTAL MODELLING ENVIRONMENT VERSION 1.0: A FRAMEWORK FOR INTEGRATING LANDFORM-SPECIFIC COMPONENT MODELS IN ORDER TO SIMULATE DECADAL TO CENTENNIAL MORPHOLOGICAL CHANGES ON COMPLEX COASTS " (ref: gmd-2016-264) with the minor revisions described below.

The readability of figure 13 has been improved by making the arrows more visible and improving the figure caption.

Should you need to contact me, please use the address indicated above.

Yours faithfully,

Andres Payo.

[Figure]

[Figure]

INVESTOR IN PEOPLE